# LiveClin: A *Live* Clinical Benchmark without Leakage

**Xidong Wang**[1,2,*] **Shuqi Guo**[1], **Yue Shen**[2], **Junying Chen**[1], **Jian Wang**[2], **Jinjie Gu**[2]
**Ping Zhang**[4], **Lei Liu**[2,3,†], **Benyou Wang**[1,†]
[1] The Chinese University of Hong Kong, ShenZhen [2] Ant Healthcare, Ant Group
[3] Zhejiang University [4] The Ohio State University

## Abstract

The reliability of medical LLM evaluation is critically undermined by data contamination and knowledge obsolescence, leading to inflated scores on static benchmarks. To address these challenges, we introduce *LiveClin*, a live benchmark designed for the approximating real-world clinical practice. Built from contemporary, peer-reviewed case reports and updated biannually, LiveClin ensures clinical currency and resists data contamination. Using a verified AI–human workflow involving 239 physicians, we transform authentic patient cases into complex, multimodal evaluation scenarios that span the entire clinical pathway. The benchmark currently comprises 1,407 case reports and 6,605 questions. Our evaluation of 26 models on LiveClin reveals the profound difficulty of these real-world scenarios, with the top-performing model achieving a Case Accuracy of just 35.7%. In benchmarking against human experts, Chief Physicians achieved the highest accuracy, followed closely by Attending Physicians, with both surpassing most models. Live-Clin thus provides a continuously evolving, clinically grounded framework to guide the development of medical LLMs towards closing this gap and achieving greater reliability and real-world utility. Our data and code are publicly available at https://github.com/AQ-MedAI/LiveClin

## 1 Introduction

Large language models (LLMs) hold immense promise for transforming healthcare, from aiding in complex diagnostics to personalizing patient care. However, the safe and effective integration of these powerful tools into clinical practice is entirely dependent on our ability to rigorously evaluate their true capabilities. As the gap between general knowledge and expert-level clinical reasoning widens, the development of sophisticated, clinically-grounded benchmarks becomes not just a matter of academic progress, but a prerequisite for building trustworthy medical AI.

However, the prevailing evaluation landscape fails to mirror clinical practice, suffering from two limitations. First, the static design of benchmarks like MedQA (Jin et al., 2020) not only makes them inherently vulnerable to data contamination and knowledge obsolescence (Oren et al., 2024; White et al., 2025) but also risks creating an illusion of capability through inflated scores. Second, their single-turn assessments are misaligned with the longitudinal nature of patient care. By evaluating reasoning in isolated, synthetic snapshots, even advanced systems like MedXpertQA (Zuo et al., 2025) and AgentClinic (Schmidgall et al., 2025) reduce patient management to a series of disconnected tasks. This approach fails to assess the integrated reasoning required to navigate a patient's entire clinical pathway, from initial presentation to long-term management.

To overcome these limitations, we introduce a clinically aligned benchmark named LiveClin, which is biannually updated and contains 1,407 clinical cases and 6,605 questions to date. Concretely, the benchmark is sourced from contemporary, peer-reviewed case reports from PubMed Central (PMC) Open Access subset to mitigate both data contamination and knowledge obsolescence. As illustrated in Figure 1, to simulate the entire clinical pathway, each case is transformed into a multi-stage exam to assess whether a model can sequentially integrate diverse modalities that reflect the patient's evolving

---

*Work done during internship at Ant Group.

Scenario: A 30-year-old African American woman presents with gradually worsening pain in her left knee that has been present for two years and has recently intensified. She denies trauma, fever, or systemic symptoms. Examination reveals focal tenderness over the proximal tibia with mild discomfort on range of motion; neurovascular status is intact and vital signs are normal.

**Stage 1: Initial Assessment**

Question: Anteroposterior radiography of the left knee is obtained (Figure 1 shows a plain X-ray). Which of the following is the most likely diagnosis suggested by the imaging findings?

Figure 1 (X-ray)

**Stage 2: Diagnostic Work-up**

Question: Magnetic resonance imaging of the knee is performed (Figure 2 shows MR sequences), and a staging chest radiograph is unremarkable (Figure 3). Which of the following is the most appropriate next diagnostic step to establish a definitive tissue diagnosis?

Figure 2 (MRI)

**Stage 3: Pathologic Diagnosis**

Question: Histologic sections from the biopsy are shown in Figure 4 (hematoxylin-eosin stain). Based on the microscopic appearance, which of the following best characterizes the lesion?

Figure 4 (Pathology)

**Stage 4: Initial Therapeutic Planning**

Question: Following confirmation of a high-grade malignant primary bone tumor, the oncology team plans neoadjuvant systemic therapy. Which regimen is considered standard first-line chemotherapy for this type of high-grade bone sarcoma?

Figure 5 (MRI)

**Stage 5: Disease Progression**

Question: Twenty months after the initial presentation of a metastatic malignant giant cell tumor (GCT) involving the spine, the patient reports severe lumbar pain radiating to the right leg. MRI of the lumbar spine is obtained (Figure 5 shows sagittal T1- and T2-weighted images). Which of the following complications is most consistent with the findings on this MRI?",

Figure 3 (X-ray)          Figure 6 (CT)

**Stage 6: Management of Refractory Metastatic Disease**

Question: Repeat thoracic imaging now demonstrates numerous bilateral pulmonary nodules (Figure 6 shows a non-contrast CT of the chest). The patient progressed despite MAP and cisplatin/doxorubicin regimens. Which of the following systemic is the most appropriate next-line therapy for her refractory metastatic disease?

Figure 1: An example from LiveClin simulating the entire clinical pathway. The case progresses from initial assessment to long-term management, with new clinical information and diverse imaging modalities (e.g., X-ray, MRI, pathology, CT) progressively introduced at each key decision point to challenge the model's reasoning in an evolving scenario.

condition. The ablation study also indicates that the overall AI-human construction workflow is superior to physician-only curation for generating challenging and high-quality content. To verify the rigor of the benchmark, we implemented a 239-physician screening pipeline, guided by the conservative principle of **rejecting any potentially flawed question**.

Our evaluation reveals a clear hierarchy in medical AI performance. Proprietary leaders such as *o3* and *GPT-5* maintain an edge but still trail Chief Physicians, achieving only marginal gains over Attending Physicians. Open-source LLMs are narrowing the gap, with large-scale models like *InternVL-3.5-241B* approaching proprietary leaders and efficient designs such as *GLM-4V-9B* surpassing weaker proprietary counterparts. And physician baselines underscore the need for targeted, domain-specific optimization, as current medical AI is approaching but has not yet achieved comprehensive management of complete clinical pathways.

**Contributions** The main contributions of this work are threefold: (1) **LiveClin**, a novel, dynamic, and multimodal benchmark that evaluates the full clinical pathway, designed to be contamination-resistant and continuously updated; (2) A scalable and verified **AI-human workflow** for generating and maintaining high-quality evaluations that mirror the clinical practice, proven to be more cost-effective and to produce more challenging questions than human-only authoring; and (3) A comprehensive **evaluation of 26 leading LLMs**, providing a new baseline for state-of-the-art clinical reasoning and uncovering critical, distinct failure modes that inform future model development.

## 2 MOTIVATION

Data contamination poses a fundamental threat to medical LLM evaluation by eroding benchmark reliability. As models are trained on ever-expanding, web-scale corpora, the questions and answers from popular static benchmarks are inevitably absorbed into their training sets (Deng et al., 2024). This widespread contamination means that models are increasingly being tested on data they have already seen, leading to inflated performance scores (Balunović et al., 2025). This is not a minor flaw but a fundamental threat to evaluation integrity, as it erodes community's ability to distinguish genuine progress from mere benchmark hacking.

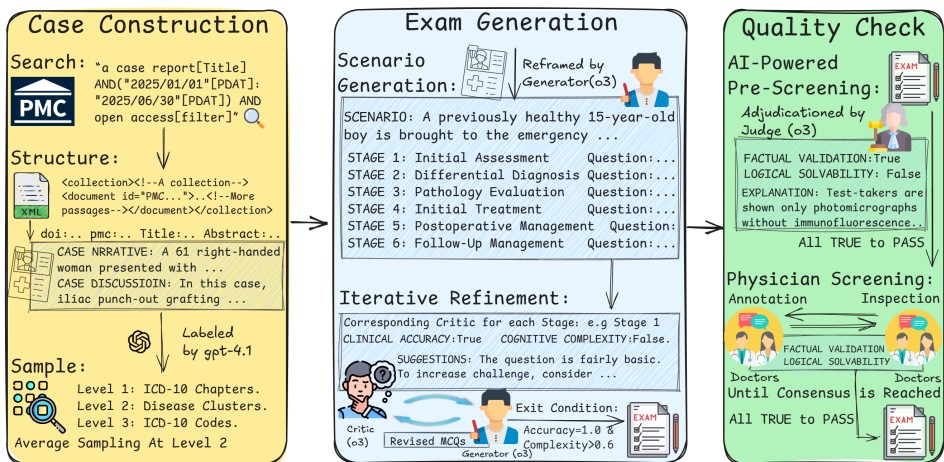

Figure 3: Overview of the LiveClin Construction Pipeline. Our three-stage pipeline creates a dynamic and clinically authentic benchmark. (1) Case Construction: We source and sample recent, peer-reviewed case reports to build a contemporary data foundation. (2) Exam Generation: An iterative Generator-Critic architecture transforms static reports into sequential reasoning problems. (3) Quality Check: A synergistic AI-human workflow, featuring AI pre-screening and multi-physician verification, ensures factual and logical integrity.

While common, decontamination efforts are merely reactive and often incomplete (Zhu et al., 2024). A truly robust solution must be proactive, designed with inherent resistance to contamination. This challenge is compounded by a second, related issue: knowledge obsolescence. Since clinical medicine evolves constantly (Cullen et al., 2019; Mitchell et al., 2023), any static test not only becomes a predictable target for contamination but also inevitably loses its clinical relevance over time.

**Pilot Study: Quantifying the Impact of Data Recency.** To empirically quantify the dual impacts of data contamination and knowledge obsolescence, we conducted a longitudinal pilot study using our main pipeline, with full methodological details available in Sections 3,4 and Appendix C. As shown in Figure 2, the results demonstrate a significant performance gap between a model's performance on older, potentially contaminated data versus novel, contemporary data. This is starkly illustrated by *GPT-5*, which scores as high as 45.0% on data within its knowledge base but drops by nearly 10 percentage points on cases published after its cutoff. This pat-

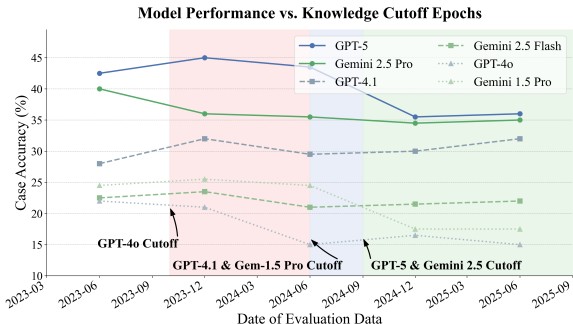

Figure 2: LLM Case Accuracy across datasets with different publication dates. Shaded regions indicate the knowledge cutoff for different models

tern, consistent across models, quantifies the distorting effects of data contamination, which inflates scores on seen data, and knowledge obsolescence, which causes failures on unseen, newer knowledge. These findings suggest that static benchmarks are an unreliable proxy for true clinical reasoning, highlighting the necessity of a live evaluation paradigm.

## 3 LIVECLIN

To address the critical need for a dynamic, contamination-resistant benchmark established in Section 2, we developed LiveClin through a rigorous methodology, overviewed in Figure 3. This section first introduces the Taxonomy (§3.1) we designed to structure the benchmark for fine-grained analysis. We then detail the three core pipeline stages: (1) Case Construction (§3.2), a process of case curation and sampling to establish a contemporary data foundation; (2) Exam Generation (§3.3), which transforms static reports into multi-step problems simulating the entire clinical pathway; and (3) Quality Check

(§3.4), a rigorous, multi-layered assurance process that guarantees the benchmark's medical validity. Finally, we present the Benchmark Statistics and Composition (§3.5).

## 3.1 Clinical Taxonomy

LiveClin's Taxonomy is a foundational framework for multi-resolution performance analysis, designed to overcome the single-score, narrow-scope limitations of existing benchmarks. It features a three-level hierarchy that evaluates model capabilities across a spectrum of granularity. The full taxonomy is detailed in Appendix D, and the analytical purpose of each level is outlined below:

- **Level 1: ICD-10 Chapters.** The highest level of our taxonomy is adapted from the authoritative ICD-10 framework to ensure its clinical and scientific validity. It consists of 16 clinically coherent chapters designed to provide a macro-level view of model capabilities across major medical specialties. To enhance focus and relevance, we merge closely related domains (e.g., Pregnancy and Perinatal Conditions) and excluded non-specific chapters (e.g., "Symptoms & Laboratory Signs") whose cases are more appropriately classified under a primary disease.

- **Level 2: Disease Clusters.** This intermediate level is adapted from the authoritative China National Healthcare Security Administration (NHSA) standard[1] to ensure a scientifically grounded framework for sub-specialty analysis. It defines 72 distinct disease clusters, balancing the need for specificity with statistical reliability. To resolve significant data sparsity encountered with the original standard, this final number was reached through an expert-guided consolidation process: merging analogous sparse groups while retaining medically unique ones to preserve specificity.

- **Level 3: ICD-10 Codes.** This most granular tier, defined by individual ICD-10 codes, enables fine-grained, diagnostic-level assessment. It is critical for identifying a model's specific strengths and weaknesses across numerous conditions. This level of detail, obscured in broader categories, gives developers the specific feedback required to improve their models and datasets.

## 3.2 Stage 1: Case Construction

With the taxonomic framework established, the first stage of our pipeline focuses on building a high-quality, structured corpus of contemporary clinical cases. This stage directly counters the challenges of **knowledge obsolescence** and **narrow disease scope** identified in existing benchmarks.

**Case Curation** The process begins by programmatically retrieving all XML-formatted case reports published in the first half of 2025 from the PubMed Central (PMC) **Open Access** Subset. A custom-built pipeline then parses each file, extracting key metadata before analyzing the article's structure. Sections describing the patient's journey (e.g., *Case Presentation*) are aggregated to form the core *case narrative*, while sections containing author analysis (e.g., *Discussion*) are consolidated into the *case discussion*. To enable multimodal proficiency assessment, this process also converts all tabular data into Markdown and extracts persistent URLs for all associated figures with their captions.

**Sampling** To construct a balanced and representative corpus, we first classify each case report against all three tiers of our taxonomy using `gpt-4.1-2025-04-14`, with detailed methodology and validation presented in Appendix E. With cases fully classified, we then implement a stratified sampling protocol guided by the 72 Level-2 disease clusters. Our protocol aims to sample 30 unique cases per cluster, while prioritizing the diversity of unique Level-3 diseases within each sample to mitigate the overrepresentation of common conditions. This rigorous procedure yielded a final corpus of 2,150 high-quality case reports, which served as the foundation for the subsequent stages.

## 3.3 Stage 2: Exam Generation

With the curated cases as our foundation, this stage focuses on generating questions that mirror the **entire clinical pathway**, moving beyond static, single-point assessments. While the challenge is achieving this at scale without sacrificing clinical nuance, we resolve the tension between manual quality and automated scalability by employing a Generator-Critic architecture, whose effectiveness is validated in Section 5. This `o3`-powered, two-agent system is guided by distinct prompts detailed in Appendix F and transforms each case report into a high-quality simulation of the patient journey.

---

[1]`https://code.nhsa.gov.cn/search.html?sysflag=8`

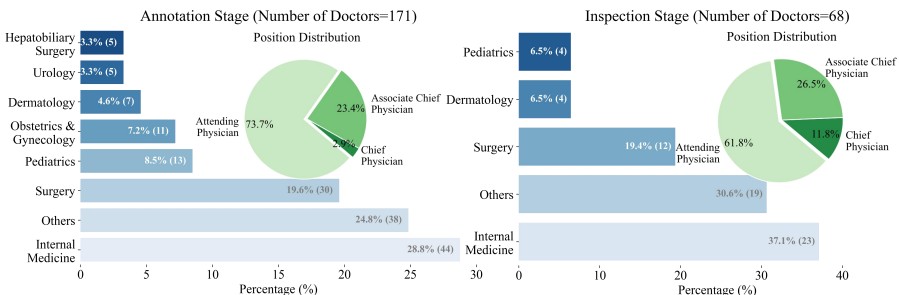

Figure 4: Distribution of the physician reviewers involved in the quality assurance process. The charts detail the team's composition by clinical specialty and professional rank (Chief Physician, Associate Chief Physician, and Attending Physician) for both the Annotation and Inspection stages.

**Scenario Generation** This process is initiated by the *Generator* Agent, which reframes each case into a progressive clinical challenge. It begins by crafting a concise initial clinical scenario capturing only the information available upon patient arrival. Building on this, the agent generates a sequence of 3–6 progressive, 10-option MCQs. To make the clinical progression explicit, the agent dynamically assigns each question a *clinical stage* label (e.g., "Initial Assessment"), ensuring a logical flow from diagnosis to long-term management. Each question's context strategically introduces new clinical details at the appropriate workflow step, probing the model's ability to integrate evolving information.

**Iterative Refinement** The centerpiece of our methodology is the closed-loop quality control orchestrated by the *Critic* Agent. Once the *Generator* produces a question set, it enters an automated "peer-review" cycle where the *Critic* evaluates it on two key dimensions: ***Clinical Accuracy*** and ***Cognitive Complexity***. If a question is flagged, the *Critic* provides actionable feedback, prompting the *Generator* to revise the set. This refinement loop persists until the question set achieves two criteria: 100% Clinical Accuracy, ensuring all content is factually correct, and high Cognitive Complexity for over 60% of its questions. To ensure efficiency, any set failing to converge within 10 cycles is discarded. Applying this process to the 2,150 curated cases yielded 2,092 high-quality question sets.

## 3.4 STAGE 3: QUALITY CHECK

Following AI-powered generation, we implement a multi-layered quality assurance protocol engineered to meet the uncompromising standards of medicine. This stage is governed by a principle of conservatism: **any question with a potential flaw is rejected**. The protocol combines AI pre-screening with multi-tiered physician verification. All evaluators apply two strict criteria: ***Factual Validation***, ensuring perfect alignment with the source case, and ***Logical Solvability***, confirming the answer is deducible from the available information. The specific prompts are detailed in Appendix G.

**AI-Powered Pre-Screening** Each generated question set first undergoes adjudication by a *Judge* Agent, implemented using o3, which acts as a highly conservative pre-filter. Systematically applying both checks by differentiating privileged from test-taker-visible information, its primary objective is to autonomously reject fundamentally flawed questions. As validated in Section 5, this AI-provided audit also serves to elevate the rigor of the subsequent physician review. This process streamlines the expert review, ultimately narrowing the pool from 2,092 to 1,869 high-potential candidates.

**Physician Screening** Our quality assurance relies on a two-phase verification by **239** licensed physicians. After AI pre-screening, cases enter an *Annotation* phase where attending physicians from top-tier hospitals assess each question against defined criteria. In the subsequent *Inspection* phase, senior physicians review the annotations. Any discrepancy initiates a revision loop with the annotator until consensus is reached. Disagreements occur in only 8.7% of cases and are fully resolved within two loops, indicating high inter-physician agreement with details shown in Appendix I.1.

As shown in Figure 4, the team combines broad specialty coverage with deep expertise, with senior experts (Chief and Associate Chief Physicians) comprising 26.3% of the Annotation team and 38.2% of the Inspection team. All major specialties and subspecialties are represented, and each case is assigned to reviewers whose expertise matches the disease category to ensure domain-appropriate evaluation, with the distribution given in Appendix I.2. All materials were translated for native-Chinese experts via o3, with translation accuracy reported in Appendix H. The process required **1,772.18 person-hours** at $24 per hour, totaling **$42,304.39**, and produced 1,822 validated question sets. From

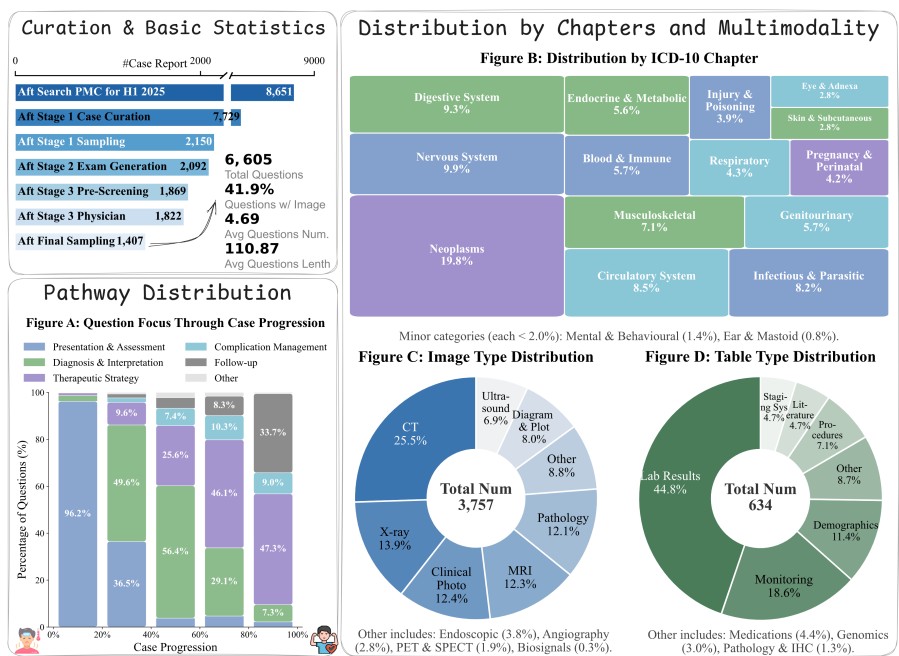

Figure 5: Overview of LiveClin's composition and statistics. The **top-left panel** details the case attrition funnel from construction pipeline and presents summary statistics of the final benchmark. **Figure A** shows the distribution of question focus across the entire clinical pathway. **Figure B** displays the distribution by ICD-10 chapters. **Figures C and D** detail the distribution of image and table modalities, respectively, highlighting the benchmark's multimodal nature.

this pool, the final benchmark of 1,407 sets was constructed using stratified sampling, selecting 20 cases per Level-2 cluster while prioritizing Level-3 disease diversity.

## 3.5 BENCHMARK STATISTICS AND COMPOSITION

The final LiveClin benchmark comprises 6,605 questions from 1,407 unique clinical cases, averaging 4.69 questions per case to form a coherent narrative. This section provides a statistical overview of the benchmark's key properties, summarized in Figure 5,

**Distribution by Clinical Pathway** A key feature of LiveClin is its simulation of the entire clinical pathway. As illustrated in Figure 5A, the question focus dynamically shifts as a case progresses to test a model's ability to handle the entire patient journey. Questions begin overwhelmingly centered on *Presentation & Assessment* (96.2% in the first quintile of a case). As more information becomes available, the focus transitions to *Diagnosis & Interpretation* and *Therapeutic Strategy* in the mid-stages. Towards the conclusion, questions concerning *Follow-up* and *Complication Management* become prominent, completing the simulation of a realistic clinical pathway.

**Distribution by Chapters and Multimodality** LiveClin spans 16 ICD-10 chapters (Figure 5B), led by *Neoplasms* (19.8%), *Nervous System* (9.9%), and *Digestive System* (9.3%). All 1,407 cases contain multimodal information, and 41.9% of questions require direct interpretation, highlighting the benchmark's emphasis on realistic clinical complexity. The dataset includes 3,757 images and 634 tables, covering medical imaging modalities (e.g., *CT*, *X-ray*, *MRI*) and structured data types (e.g., *Lab results*, *Monitoring*, *Demographics*) as shown in Figures 5C and 5D. Stage-specific analysis shows multimodal content is most frequent in early workflow stages, less common in mid-stages, and increases again toward the final stages, reflecting shifts in diagnostic and management priorities. Detailed stage-level results are provided in Appendix L.

## 4 EXPERIMENTS

In this section, we present a comprehensive evaluation of leading large language models (LLMs) on the LiveClin benchmark to assess their clinical reasoning capabilities. We first detail the Experimental

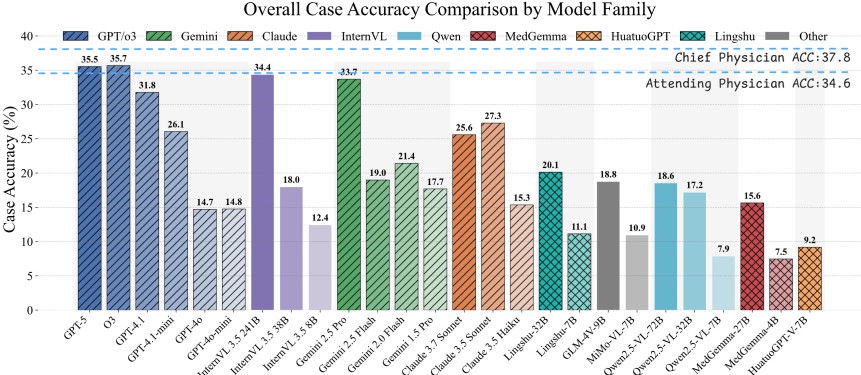

Figure 6: Overall case accuracy, showing models grouped by family and ordered reverse chronologically. Bar textures indicate model type and dashed lines represent physician reference levels.

Setup (§4.1), including the models tested and our evaluation protocol. We then report the Overall Performance Evaluation (§4.2), providing a high-level comparison of model capabilities. Following this, an In-depth Analysis (§4.3) explores model performance across different clinical domains and modalities, leveraging the fine-grained structure of LiveClin.

## 4.1 EXPERIMENTAL SETUP

**Evaluated Models and Human Experts** We conduct evaluation on a comprehensive set of 26 models. Our selection spans three key categories: proprietary models like *GPT-5* (OpenAI et al., 2024), powerful open-source general LLMs like *Qwen2.5-VL* (Bai et al., 2025), and medical-specific LLMs like *MedGemma* (Sellergren et al., 2025). The complete list of all model versions is available in Appendix M. In addition to models, we benchmarked physicians on 100 randomly sampled LiveClin cases, recording accuracy separately for Chief and Attending Physicians. The assessment mirrored the model protocol for consistency, with details in Appendix M.2.

**Evaluation Protocol and Metrics** To faithfully simulate sequential clinical encounters, we employed a conversational, zero-shot evaluation protocol. The full conversation history is maintained as context for each subsequent question, forcing the model to continuously integrate new information. For reproducibility, we set `temperature` to 0 for most models, adopting official recommended configurations for those with specific reasoning modes. Our primary metric is Case Accuracy, a stringent measure where a case is deemed correct only if *all* of its sequential questions are answered correctly. Further details on our prompting strategy and robustness analyses of the evaluation paradigm are presented in Appendix M.1 and M.3 separately.

## 4.2 OVERALL PERFORMANCE EVALUATION

As shown in Figure 6, overall Case Accuracy across leading LLMs offers a clear view of the current state of clinical AI, revealing both advanced capabilities and persistent challenges.

**Present Hierarchy: From Human Experts to Rising Stars** Proprietary models lead the field, with *o3* and *GPT-5* at the top. Benchmarking against physicians on 100 randomly sampled LiveClin cases shows Chief Physicians with the highest accuracy, Attending Physicians slightly lower, and both groups outperforming most models. Only *GPT-5* and *o3* marginally exceeded Attending Physicians yet still fell short of Chief Physicians. This gap between leading models and experienced clinicians highlights the benchmark's difficulty, alongside the spread between top-tier systems and smaller models such as *GPT-4o-mini*. Open-source models are narrowing the gap, with the large-scale *InternVL-3.5-241B* approaching proprietary leaders, and efficient designs such as *GLM-4V-9B* surpassing weaker proprietary systems like *GPT-4o*.

**Future Trajectory: Closing the Gap in Clinical Expertise** Our findings challenge the belief that scaling or newer releases alone deliver better clinical reasoning. For example, *Claude 3.5 Sonnet* outperforms its successor *Claude 3.7 Sonnet*, and within Gemini, *Gemini 2.0 Flash* scores higher than *Gemini 2.5 Flash*. This signals the end of automatic gains from general upgrades and points to the need for targeted, domain-specific optimization. Physician baselines strengthen this view: despite recent improvements in medical-specific models like *Lingshu-32B*, a clear gap remains compared

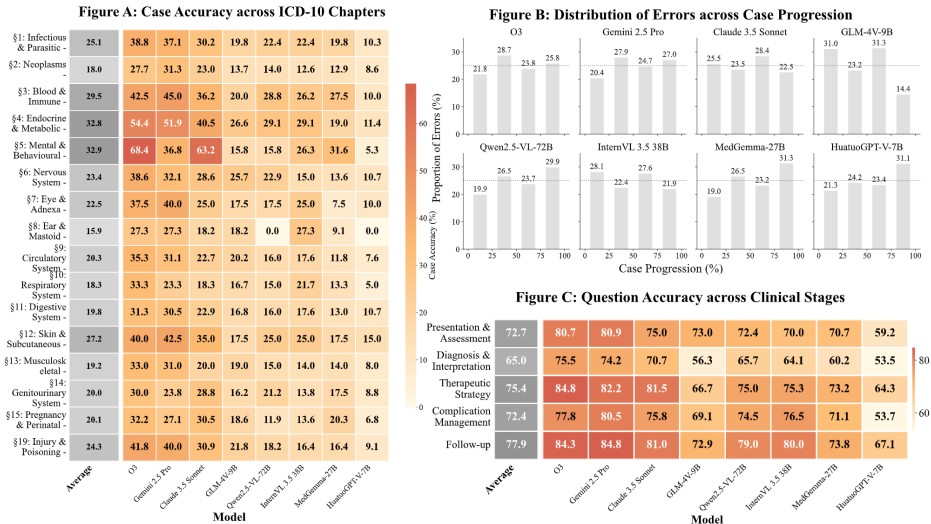

Figure 7: **Fine-grained Performance Analysis of Representative Models. Figure A** shows Case accuracy (%) across 16 major ICD-10 chapters. **Figure B** details Distribution of error proportion (%) across case progression. **Figure C** displays Question accuracy (%) across five clinical stages.

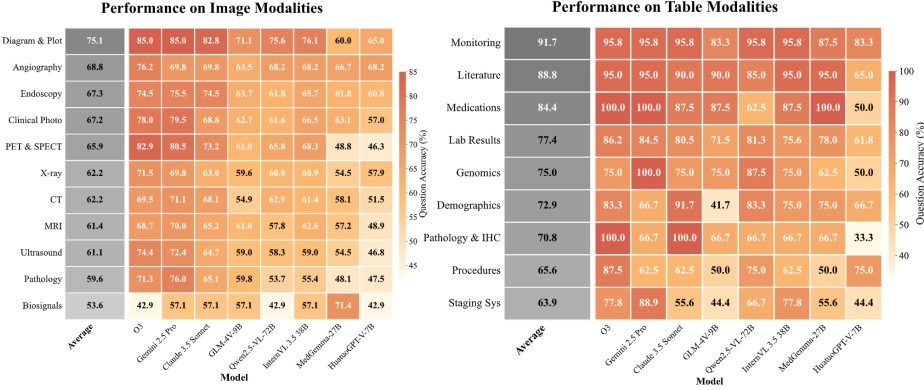

Figure 8: **Performance Analysis across Multimodal Sub-types.** Question accuracy (%) on distinct image and table modalities. Modality sub-types are sorted by the average accuracy across all models.

with Chief Physicians, showing that current medical AI is far from managing complete clinical pathways. Building on stronger generalist foundations and aligning with expert-level reasoning will be essential to close this gap.

## 4.3 IN-DEPTH ANALYSIS

To dissect model capabilities beyond aggregate scores, we conducted a fine-grained analysis of eight representative models across clinical domains, the cilinical pathway, and data modalities. This reveals not just what models get wrong, but how and when their reasoning fails.

**Errors in Motion: Failure Modes Along the Pathway.** An analysis of error patterns across the clinical pathway reveals distinct failure modes characteristic of different model classes (Figure 7B-C). Top proprietary models like *o3* tend to fail mid-pathway, with errors peaking during the cognitively demanding Diagnosis & Interpretation phase. In contrast, open-source medical models exhibit the late-stage failure pattern, where errors cluster in the final quartile during the less complex Follow-up stage, suggesting a critical breakdown in long-context retention. Finally, general-purpose models such as *GLM-4V-9B* exhibit a front-loaded error profile, stumbling early in the process. This highlights an urgent need to improve their ability to reason effectively from the initial clinical presentation.

**Errors in Place: Strengths and Weaknesses in Domains.** Our analysis of ICD-10 chapters reveals that model performance is highly variable, uncovering distinct specializations alongside universal

| Method | Accuracy (%) | Trivial Ratio (%) | Time (hrs)[†] | Cost ($) |
|---|---|---|---|---|
| Physicians | 92.5 | 38.5 | 188.9 | 4534.30 |
| Generator | 84.5 | 16.5 | 0.13 | 35.34 |
| Generator-Critic | 93.0 | 5.5 | 0.45 | 221.69 |
| Generator-Critic-Judge | 89.5[*] | 5.5 | 0.55 | 244.19 |

Table 1: Ablation study of exam generation methods. Accuracy is the physician-validated pass rate. Costs are normalized per 100 case reports. [†]Time denotes person-hours for human authors or API run-time for AI. [*]The accuracy on the Judge-approved subset is 98.4%.

weaknesses (Figure 7A). For instance, models excel in areas governed by clear systemic logic, such as Endocrine Diseases, yet falter universally in domains that demand nuanced synthesis, like Neoplasms. Interestingly, this specialization transcends scale: both the top-tier *o3* (68.4%) and the compact *Claude-3.5-Sonnet* (63.2%) achieve exceptional accuracy in Mental & Behavioural Disorders.

This pattern of domain-specific proficiency extends to multimodal reasoning, where a critical gap separates simple data extraction from complex inference (Figure 8). Models confidently interpret structured data like Diagrams (75.1%) but struggle when expert-level reasoning is required, evidenced by poor performance on modalities such as Pathology (59.6%) and Biosignals (53.6%). Although specialized training shows promise, with *MedGemma-27B* displaying a surprising aptitude for Biosignals (71.4%), foundational robustness remains a key challenge. Even the most capable models can falter on seemingly simple inputs like Demographics tables, underscoring this core issue.

## 5 ABLATION STUDY OF THE AGENT WORKFLOW

To validate the contribution of each component in our agent-based pipeline, we conducted an ablation study on randomly sampled 200 case reports. We benchmarked our full *Generator-Critic-Judge* pipeline against three alternatives: a physician-authored baseline, a Generator-only model, and a Generator-Critic pipeline. Each configuration was evaluated for accuracy, cost, and complexity. We quantified complexity as the proportion of 'trivial' questions, which are defined as questions correctly solved by all three baseline models (GPT-4o-mini, Claude-3.5 Haiku, Gemini-1.5 Pro). The detailed comparison across all evaluation metrics is presented in Table 1.

**Automated Exam Creation: Scaling with *Generator* and *Critic*.** LLM-based generation substantially improved both scalability and question complexity. Operating alone, the *Generator* agent reduced time and financial costs by nearly two orders of magnitude compared to physician authoring, enabling continuous updates, while lowering the proportion of trivial questions from 38.5% to 16.5%. Adding the *Critic* agent was essential for factual accuracy, raising physician-validated accuracy from 84.5% to 93.0% and further reducing the trivial ratio to 5.5%. This iterative refinement is critical for producing reliable, clinically demanding content at scale. To assess whether the increased difficulty reflected genuine clinical complexity rather than artifacts, we conducted an attribution analysis. Over 80% of reclassified cases were attributed to clinically relevant factors, chiefly the retention of authentic but incomplete information, requirements for cross-stage or multi-step reasoning, and refinement of distractor plausibility to mirror realistic differential diagnoses. Full methodological details and category-level results are provided in Appendix N.

**Augmented Quality Check: Enhancing Verification with *Judge*.** The final *Judge* agent functions not merely as a filter but as a crucial enhancement to the human review process. Although its inclusion nominally lowers the pass rate to 89.5%, this decrease signifies a positive outcome: a more rigorous quality standard. By providing physicians with a structured audit trail and direct evidence from the source case, the *Judge* enables them to identify subtle flaws that might otherwise be overlooked. By providing a structured audit trail and highlighting potential issues, *Judge* empowers physicians to conduct their validation with efficiency and rigor.

## 6 SUSTAINABILITY, CONTAMINATION CONTROL, AND BIAS EVALUATION

Beyond validating the agent workflow, LiveClin's long-term reliability rests on sustainability, contamination control, and bias evaluation. Below we present the strategies and evidence for each.

| Stage I | Stage II | Stage III | Evaluation | Total (Cost/Time) |
|---|---|---|---|---|
| 0 / 0.8 | 3,500 / 0.5 | 45,000 / 9.0 | 5,000 / 1.0 | 53,500 / 11.3 |

Table 2: Estimated cost in *USD* and time in *Days* required for each stage of a biannual update cycle of LiveClin. Values are reported as `Cost/Time`.

| Time | GPT-5 | GPT-4.1 | InternVL 3.5-24B | Gemini 2.5 Pro | Claude 3.5 Sonnet | Qwen2.5-VL-72B | MedGemm 27B | LingShu-32B | HuatuoGPT-V-7B |
|---|---|---|---|---|---|---|---|---|---|
| H1 2025 | 35.5 | 31.8 | 34.4 | 33.7 | 27.3 | 17.2 | 15.6 | 20.1 | 9.2 |
| July 2025 | 36.8 | 33.2 | 35.8 | 35.1 | 28.7 | 18.6 | 16.9 | 21.4 | 10.5 |
| Aug 2025 | 34.1 | 30.4 | 33.0 | 32.3 | 26.0 | 15.8 | 14.2 | 18.7 | 8.0 |

Table 3: Monthly stability evaluation of the private leaderboard, showing Case Accuracy (%) for each model across the first half of 2025 (H1 2025), July 2025, and August 2025 sets.

**Sustainability of Biannual Updates** We maintain a physician-reviewed update cycle every January and July as a core requirement for reliable live medical AI evaluation. Each cycle replaces the entire evaluation set, reassesses existing models, and includes newly released ones. Leveraging our AI–human collaborative workflow, cases from the preceding six months are collected, validated, and published within the first two weeks. The estimated resource demands in Table 2 indicate that this schedule is sustainable within our current capacity.

**Contamination Control and Monitoring** We implement periodic updates to limit contamination risk, following practices in LiveBench (White et al., 2024) and LiveCodeBench (Jain et al., 2024). A lag of approximately six to eight months between model data collection and public release provides an effective window for contamination control. To detect potential exploitation from frequent iterations by individual developers, we operate a private leaderboard updated monthly, with its construction and evaluation details presented in the Appendix O. As shown in Table 3, monthly score variations are small and rankings remain stable, confirming that our monitoring safeguards benchmark integrity.

**Bias Considerations and Impact Assessment** Patient populations differ substantially across regions, healthcare tiers, and specialties, precluding the definition of a universal gold-standard benchmark distribution. Case reports typically present unexpected findings, novel pathogenic insights, and innovative therapeutic approaches, aligning with our goal of capturing emerging medical knowledge through continuous updates. However, case reports contain a higher proportion of rare conditions, which could bias evaluation. To quantify this effect, each case is annotated by clinicians as rare or common, and model performance is compared across subsets. As detailed in Appendix P, accuracy differences are within five percentage points for most models and rankings are largely preserved. Stronger models are less affected by rarity, with *GPT-5*, *o3* and *GPT-4.1* even perform better on the rare subset, suggesting that rarity has limited impact on scores and rankings.

We also acknowledge that PubMed underrepresents low-resource regions, reflecting global publication patterns, language barriers, and indexing criteria. Closing this gap requires coordinated effort across the research community to expand data from underrepresented settings. For future work, the community needs to enhance coverage by sourcing cases from regional repositories, non-English literature, and clinician networks to build more diverse and globally representative evaluations.

## 7 CONCLUSION

To combat the threats of data contamination and knowledge obsolescence in medical LLM evaluation, we introduced **LiveClin**, a dynamic benchmark built from a constantly refreshed stream of contemporary, peer-reviewed case reports. Our AI-powered workflow transforms these cases into multimodal challenges that span the entire clinical pathway. Evaluation on LiveClin reveals a stark performance gap, with a top Case Accuracy of just 35.7%, and uncovers distinct failure modes across the clinical journey, such as mid-case synthesis struggles in top models and late-stage context loss in specialized ones. LiveClin marks a paradigm shift from static knowledge testing to the dynamic assessment of applied clinical reasoning. By providing a continuously evolving and clinically-grounded challenge, we aim to guide the development of medical LLMs towards greater real-world reliability and safety.

## ACKNOWLEDGMENTS

B.W was supported by Shenzhen Medical Research Fund (B2503005)，Major Frontier Exploration Program (Grant No. C10120250085) from the Shenzhen Medical Academy of Research and Translation (SMART), the Shenzhen Science and Technology Program (JCYJ20220818103001002), NSFC grant 72495131, Shenzhen Doctoral Startup Funding (RCBS20221008093330065), Tianyuan Fund for Mathematics of National Natural Science Foundation of China (NSFC) (12326608), Shenzhen Science and Technology Program (Shenzhen Key Laboratory Grant No. ZDSYS20230626091302006), the 1+1+1 CUHK-CUHK(SZ)-GDSTC Joint Collaboration Fund, Guangdong Provincial Key Laboratory of Mathematical Foundations for Artificial Intelligence (2023B1212010001), and Shenzhen Stability Science Program 2023. P.Z. is not funded by any of these funders.

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

## A  Use of Large Language Models (LLMs)

To ensure the linguistic accuracy and fluency of this manuscript, a Large Language Model (LLM) was employed as a writing-enhancement tool. The use of the LLM was focused on three areas:

- **Grammar and Spell Checking:** Identifying and correcting grammatical errors, typos, and punctuation mistakes.

- **Wording and Phrasing Refinement:** Optimizing word choice and sentence structure to enhance clarity, flow, and academic tone.

- **Data Construction:** Constructing Benchmark as Agent.

## B  Related Work

**Medical Datasets**  The evaluation of medical LLMs has rapidly evolved to address the complexities of clinical practice. Initial efforts focused on static, text-only question-answering datasets such as MedQA (Jin et al., 2020), MedMCQA (Pal et al., 2022) and the medical subsets of MMLU (Hendrycks et al., 2021). While foundational, these benchmarks primarily test static knowledge recall. To incorporate visual data, multimodal resources including VQA-RAD (Lau et al., 2018) and PathVQA (He et al., 2020) were developed, with more recent efforts such as MedXpertQA (Zuo et al., 2025) expanding difficulty and clinical context. However, these approaches largely retain a single-turn, static format. A parallel line of work has explored multi-step evaluation. Agent-based benchmarks such as AgentClinic (Schmidgall et al., 2025) and MedAgentBench (Jiang et al., 2025) simulate clinical workflows though often with synthetic scenarios. NEJM CPC benchmark (McDuff et al., 2023) evaluates multimodal diagnostic reasoning with multi-stage pathways based on case reports from the New England Journal of Medicine, yet relies on historical data without contamination safeguards. MedCaseReasoning (Wu et al., 2025) focuses on medical reasoning from clinical case reports but remains text-only and single-turn in structure. CaseReportBench (Zhang et al., 2025) supports dense information extraction from clinical case reports but lacks multi-stage pathway simulation and temporal robustness. A distinct approach is presented by HealthBench (Arora et al., 2025), which evaluates open-ended, multi-turn conversations using thousands of physician-designed rubric criteria to assess not only accuracy but also safety and communication quality. CMB (Wang et al., 2024) has highlighted the importance of localization. LiveClin complements these diverse efforts by tackling two critical, unaddressed dimensions: the temporal decay of medical knowledge that makes static benchmarks quickly obsolete, and the evaluation of sequential reasoning grounded in authentic, contemporary patient cases with explicit long-tail disease coverage.

**Data Contamination and Live Benchmarks**  The vulnerability of static benchmarks to data contamination is a well-documented problem, as models can achieve inflated scores by memorizing test items seen during pretraining rather than demonstrating true reasoning capabilities (Oren et al., 2024; Zhou et al., 2023). This risk is particularly acute in high-stakes medical contexts, where the leakage of data from sources like licensing exams can create a false sense of security about a model's real-world clinical utility (Kung et al., 2023; Zhang et al., 2024). To address these challenges, dynamic or "live" evaluation paradigms have emerged as a robust solution, designed to counter contamination by continuously introducing unseen, time-stamped data. This approach has been successfully implemented across diverse domains. LiveBench (White et al., 2025) established a foundational model by automatically sourcing new questions from recent public data, which was extended to specialized fields with LiveCodeBench (Jain et al., 2024) using challenges from recent coding contests. Pushing the paradigm further, FutureX (Zeng et al., 2025) introduced a live benchmark for agent-based future prediction, a task requiring real-time information retrieval. While these pioneering efforts effectively mitigate data leakage, they remain primarily text-centric and do not address the tripartite challenge of clinical evaluation: the need to integrate diverse **multimodal** data, assess complex **sequential reasoning** in realistic patient workflows, and ensure **comprehensive disease coverage**. LiveClin bridges this critical gap. It adapts the live paradigm to the clinical domain by grounding evaluation in a constantly refreshed corpus of authentic, peer-reviewed patient cases, and uniquely embeds temporally-staged multimodal evidence within a structured workflow to enable a dynamic assessment of clinical reasoning that spans the full ICD-10 taxonomy.

## C  Pilot Study Methodology

This appendix provides the detailed methodology for the longitudinal pilot study presented in Section 2. The following sections detail the dataset construction and evaluation protocol, highlighting key distinctions from the main LiveClin pipeline that were made to specifically isolate the impact of data recency in a cost-effective manner.

**Dataset Construction**  The construction process began with the random sampling of an initial pool of case reports from the PubMed Central (PMC) Open Access Subset for five distinct biannual periods (2023-S1 to 2025-S1). This initial sampling strategy differs from the stratified approach used for the main benchmark, as the goal here was to capture a typical temporal cross-section rather than a taxonomically balanced one.

These cases were then processed through the AI-powered generation and refinement pipeline described in Section 3, utilizing both the `Generator-Critic` architecture for question creation and the `Judge` Agent for automated quality screening. In a key departure from our main pipeline and to ensure cost-efficiency for this preliminary study, the multi-tiered physician verification phase was omitted. This AI-only pipeline resulted in a variable number of successfully generated question sets for each temporal slice. To ensure a fair and balanced comparison across periods, we performed a final random sampling step to create five standardized test sets, each containing 200 question sets.

**Model Evaluation Protocol**  For the evaluation phase of the pilot study, we selected a representative set of leading proprietary models, as these are often the subject of contamination discussions and their performance provides a meaningful signal of the state-of-the-art. The evaluation itself was conducted under conditions identical to our main experiments to ensure the comparability of our findings. We employed the exact same sequential, zero-shot evaluation protocol detailed in Section 4 and Appendix M. This consistency in the testing framework ensures that any observed performance differences can be confidently attributed to the changing temporality of the data, not variations in the evaluation method.

## D  LiveClin Clinical Taxonomy

This appendix provides the complete details of the three-level clinical taxonomy introduced in Section 3.1. This hierarchical framework is a foundational component of LiveClin, designed to enable a multi-resolution performance analysis that moves beyond the single-score, narrow-scope evaluations of traditional benchmarks.

Table 4 details the full structure, mapping the 16 Level-1 Clinical Chapters to their constituent 72 Level-2 Disease Clusters. Level 3, the most granular tier, corresponds to the specific ICD-10 codes used for fine-grained analysis of individual conditions. The design of this taxonomy, particularly the expert-guided consolidation at Level 2, was crucial for balancing analytical specificity with the statistical reliability needed for robust model evaluation.

## E  Details for Case Report Classification

To automatically classify each case report into our three-level taxonomy, we employed `gpt-4.1-2025-04-14`. We designed a hierarchical, two-stage classification pipeline to enhance accuracy and consistency by breaking down the complex task into simpler, sequential steps. The process first identifies the broad clinical chapter (Level 1) and then, based on that result, determines the more specific disease cluster (Level 2) and ICD-10 code (Level 3).

**Stage 1: Level-1 Chapter Classification**  In the first stage, the model is provided with the case report's title and abstract, along with the complete list of 16 Level-1 chapters. Its task is to select the single most relevant chapter for the case. The prompt used for this stage is summarized in Figure 9.

**Stage 2: Level-2 Cluster and Level-3 ICD-10 Classification**  Once the Level-1 chapter is determined, the pipeline proceeds to the second stage. The model is again given the case's title and abstract, but this time it is provided with a constrained list of options containing only the Level-2 disease clusters and Level-3 ICD-10 codes that fall under the previously identified chapter. This significantly reduces the search space and improves precision. The model's task is to select the most fitting Level-2 cluster

| ICD-10 Chapters (Level 1) | Disease Clusters (Level 2) |
|---|---|
| §1: Infectious & Parasitic | Intestinal Inf. & Tuberculosis (A00-19); Zoonotic bacterial (A20-28); Bacterial, Spirochetal, Chlamydial Inf., etc. (A30-79); Viral Inf. (CNS, HIV, etc.) (A80-B34); Fungal & Protozoal Inf. (B35-64); Parasitic & Helminthic (B65-98) |
| §2: Neoplasms | Malignant Neoplasms of Lip, Oral, Pharynx (C00-14); Digestive Organs (C15-26); Respiratory & Intrathoracic (C30-39); Bone & Cartilage (C40-41); Skin (C43-44); Mesothelial & Soft Tissue (C45-49); Breast (C50); Female Genital Organs (C51-58); Male Genital Organs (C60-63); Urinary Tract (C64-68); Eye, Brain & CNS (C69-72); Thyroid & Endocrine (C73-75); Lymphoid Tissues (C76-96); Benign, In Situ. (C97, D00-48) |
| §3: Blood, Blood-forming Organs & Immune Mechanism | Nutr, Hemol & Aplast Anemias (D50-64); Coagulation Defects, Purpura & Hemorrhagic Cond. (D65-69); Blood-Forming Organ (D70-77); Immune Mechanism Disord. (D80-89) |
| §4: Endocrine, Nutritional & Metabolic | Thyroid Disorders. (E00-07); Diabetes mellitus (E10-14); Endocrine, Glucose Reg. & Nutritional Deficiencies (E15-64); Obesity & Metabolic Disorders (E65-90) |
| §5: Mental & Behavioural Disord. | Comprehensive Mental & Behavioural Disord. (F00-98) |
| §6: Nervous System | Inflammatory & systemic atrophic of CNS (G00-14); Extrapyramidal, Movement, & Degen. Disord. (G20-32); Demyelinating of CNS (G35-37); Episodic & paroxysmal disord. (G40-47); Nerve root & plexus disord. (G50-59); Polyneuropathies & other PNS disord. (G60-64); Muscle, Myoneural, & Paralytic Disord. (G70-99) |
| §7: Eye & Adnexa | Disord. of eyelid, lacrimal system, orbit, conjunctiva, sclera, etc. (H00-22); Disord. of Lens, Retina, Glaucoma, Globe, vitreous body, optic nerve, visual pathways, etc. (H25-59) |
| §8: Ear & Mastoid Process | Comprehensive Ear & Mastoid Disord. (H60-95) |
| §9: Circulatory System | Rheumatic, Hypertensive, & Ischemic Heart (I00-25); Pulmonary heart & circulation (I26-28); Other forms of heart (I30-52); Cerebrovascular (I60-69); Arteries, arterioles & capillaries (I70-79); Venous, Lymphatic (I80-99) |
| §10: Respiratory System | Acute & Chronic Respiratory Infections and Disorders (J00-39); Chronic, Environmental & Pleural (J40-94); Other diseases of the respiratory system (J95-99) |
| §11: Digestive System | Oral Cav, Saliv Glands & Jaws (K00-14); Esoph, Stomach, Appendix (K20-38); Liver (K70-77); Gallbldr, Biliary Tract, Pancreas (K80-93); Hernia (K40-46); Noninfective enteritis & colitis (K50-52); Intestinal & Peritoneal Disorders (K55-67) |
| §12: Skin & Subcutaneous Tissue | Infectious, Bullous & Eczematous Skin. (L00-30); Papulosquamous, Urticarial, Radiation & Miscellaneous Skin. (L40-99) |
| §13: Musculoskeletal & Connective Tissue | Arthropathies (M00-25); Systemic connective tissue disord. (M30-36); Dorsopathies (M40-54); Soft tissue disord. (M60-79); Osteopathies, Chondropathies, etc. (M80-99) |
| §14: Genitourinary System | Glomerular, Tubulo-Interstitial & Renal Failure Disord. (N00-19); Urolithiasis and Other Kidney & Ureter Disord. (N20-29); Urinary System & Male Genital Organ Disord. (N30-51); Breast & Female Reproductive System. (N60-99) |
| §15: Pregnancy, Childbirth & Perinatal Period | Pregnancy with Abortive Outcome & Maternal Disord. (O00-48); Labor, Delivery & Puerperal Complications (O60-92); Fetus Affected by Maternal, Perinatal Factors (P00-83) |
| §19: Injury, Poisoning & External Causes | Injuries & Foreign Bodies by Body Region (S00-T19); Burns, Frostbite, & Drug Poisoning (T20-50); Toxic Effects of Substances Chiefly Nonmedicinal as to Source (T51-65) |
| **Total** | 72 |

Table 4: Hierarchical Taxonomy of the LiveClin Benchmark. The table displays the first two levels of the taxonomy. Level 3, the most granular tier, corresponds to the specific ICD-10 codes.

---

**Prompt for Level-1 Classification**

Based on the following medical case report title and abstract, please identify the most relevant chapter from the medical classification system provided below.
*[List of all 16 Level-1 Chapters is inserted here]*
Case Report Title: *"title"* Case Report Abstract: *"abstract"*
Instructions: Provide your classification after a '—' separator line. Format the response exactly as follows:
Level1: Chapter *[Number]*: *[Chapter Name]*

---

Figure 9: The prompt designed to classify a case report into one of the 16 high-level clinical chapters.

and the most specific Level-3 ICD-10 code from these filtered lists. The core structure of this prompt is shown in Figure 10.

**Validation** To validate the reliability of this automated pipeline, we randomly sampled 200 case reports from our corpus. We then invited clinical experts to manually assign the appropriate Level-1,

> **Prompt for Level-2 and Level-3 Classification**
>
> Based on the medical case report, which belongs to *[Pre-identified Level-1 Chapter]*, please provide:
> The most specific sub-category (Level 2) from the list below.
> The most specific ICD-10 code WITH its disease name that matches this case.
> Available Level-2 categories for this chapter (YOU MUST CHOOSE ONE FROM THIS LIST):
> *[List of relevant Level-2 Disease Clusters is inserted here]*
> Allowed ICD-10 codes in this chapter (you MUST copy exactly one line from below):
> *[List of relevant Level-3 ICD-10 Codes is inserted here]*
> Case Report Title: *"title"* Case Report Abstract: *"abstract"*
> Instructions: Provide your classifications after a '—' separator line. Format EXACTLY as follows:
> Level2: *[Sub-category Name] ([Code Range])* ICD-10: *[Code] [Disease Name]*

Figure 10: The prompt for the second classification stage. It uses the Level-1 result to provide a constrained set of options for Level-2 and Level-3 classification.

Level-2, and Level-3 labels to these cases. The model's classifications were compared against these expert annotations, achieving accuracies of **98.5%** for Level-1 Chapters, **97.5%** for Level-2 Disease Clusters, and 97.0% for Level-3 ICD-10 codes. These high accuracy rates confirmed the method's suitability for constructing our benchmark.

## F   PROMPTS FOR *Generator* AND *Critic* AGENT

As described in Section 3.3, our iterative refinement pipeline is powered by a *Generator-Critic* architecture. Both agents are instances of the `o3` large language model, guided by distinct, highly-structured prompts to fulfill their specific roles. This section details the core instructions provided to each agent.

***Generator* Agent**   The *Generator*'s primary role is to transform a static, unstructured case report into a dynamic, multi-step clinical reasoning challenge. It receives the full case report text, including figure/table captions and the discussion section, as its ground-truth context. Its prompt instructs it to synthesize this information into a progressive series of Multiple-Choice Questions (MCQs) that simulate a clinical workflow. Key constraints emphasize creating plausible distractors, ensuring questions test image interpretation rather than caption recall, and strictly forbidding any reference to the source "case report," which is invisible to the test-taker. The core of this initial generation prompt is outlined in Figure 11.

***Critic* Agent**   The `Critic` agent orchestrates the quality control loop. After the `Generator` produces an initial set of MCQs, the `Critic` receives both this generated set and the original ground-truth case report. Its prompt (Figure 12) instructs it to perform a meticulous "peer review" of each MCQ. The evaluation is structured around two key dimensions: *Clinical Accuracy* (verifying factual correctness against the source report and ensuring sufficient information is present to answer) and *Cognitive Complexity* (assessing whether questions demand high-level reasoning and feature challenging distractors). The agent must output its structured feedback in a JSON format, providing specific critiques and a binary judgment for each dimension.

***ReGenerator* Phase**   If the *Critic* flags any issues, the process enters a revision phase. The *Generator* agent is invoked again, but with an expanded prompt. This "ReGenerator" prompt (Figure 13) includes the original case report, the previously generated (flawed) MCQs, and, crucially, the structured feedback from the *Critic*. The agent's task is to regenerate the entire examination set, specifically addressing every point of critique to improve the quality of the questions. This loop continues until the *Critic*'s criteria are fully met.

## G   PROMPTS FOR *Judge* AGENT AND DOCTORS

The multi-layered quality assurance protocol described in Section 3.4 is guided by a unified set of evaluation criteria applied to both AI and human reviewers. The core of this protocol is a prompt designed to enforce a strict, two-part validation for every question. This ensures that each question

---

**Prompt for `Generator` Agent**

**Goal:** Generate an initial clinical scenario and 3-6 progressive MCQs (10 options each) based *only* on the provided case report sections.

**CRITICAL CONSTRAINTS:**

- **FORBIDDEN CONTENT REFERENCES:** NEVER mention "case report," "case description," or "discussion." Test-takers only see the scenario, progressive question info, and Figures/Tables.

- **MULTIMODAL ASSESSMENT:** NEVER describe image findings in the question text. Only state the figure number and modality (e.g., "Figure 1 shows an ultrasound..."). Questions MUST test image interpretation ability.

- **DISTRACTOR QUALITY:** ALL 9 incorrect options MUST be clinically plausible distractors representing realistic differential diagnoses. They must require clinical reasoning to eliminate.

**Instructions:**

1. **Create an Initial Scenario:** Describe *only* the patient's initial presentation.

2. **Generate 3-6 Progressive MCQs:**
   - Frame questions as queries a clinician might ask an AI assistant.
   - Arrange questions along a logical clinical timeline (e.g., assessment -> diagnosis -> treatment).
   - Incrementally introduce new clinical findings from the case report within each question's stem.

3. **Output Format:** MUST output ONLY a valid JSON object containing the scenario and a list of MCQs.

---

Figure 11: The prompt for the `Generator` agent, which initiates the creation of the clinical examination by converting a case report into a progressive set of MCQs.

---

**Prompt for `Critic` Agent**

**Your Task:** You are an expert medical exam question critic. Evaluate **each of the MCQs** provided below. For each one, consider the initial scenario, all information revealed in preceding MCQs, and the ground-truth Case Report Context.

**Evaluation Criteria for EACH MCQ:**

1. **Correctness & Clarity:**
   - **Information Sufficiency:** Based *strictly* on the information available to the test-taker at this stage, is there enough detail to unambiguously arrive at the correct answer?
   - **Reference Violations:** Does the question improperly reference invisible content like the "case description"?
   - **Distractor Logic:** Can each incorrect option be eliminated with clear clinical reasoning based on the available information?

2. **Difficulty & Cognitive Level:**
   - **Challenge Level:** Is the question challenging enough for a clinician-level AI assistant?
   - **Distractor Quality:** Are the incorrect options strong enough to require complex reasoning to eliminate?

**Output Format:** For each MCQ, provide a JSON object containing two evaluations: *correctness_evaluation* and *difficulty_evaluation*. Each must include a boolean flag (*is_correct_and_clear*, *is_sufficiently_difficult*) and a detailed text field for *critique_and_suggestions*.

---

Figure 12: The prompt for the *Critic* agent, designed to systematically evaluate the generated MCQs for clinical accuracy and cognitive complexity.

is not only factually correct according to the source material but also logically solvable from the perspective of an examinee who lacks access to that privileged information.

***Judge* Agent Prompt** For the AI pre-screening step, a *Judge* agent (an instance of *o3*) is provided with the full context: the ground-truth case report and the finalized set of questions. The agent's prompt, summarized in Figure 14, explicitly instructs it to adopt two distinct personas for evaluation.

> **Prompt for *ReGenerator* Phase**
>
> **Goal:** Revise and improve a set of medical exam questions based on critiques.
> **Context:**
>
> 1. **Original Patient Case Report:** [Full text of the source case report]
> 2. **Previously Generated Exam Content:** [The full scenario and list of MCQs from the previous attempt]
> 3. **Critique and Suggestions for Improvement:** [The structured feedback from the *Critic* agent]
>
> **Your Task:** Based on all the information above, **regenerate the entire exam content (initial scenario and all MCQs)**, addressing ALL feedback provided in the "Critique and Suggestions." Your primary focus is to fix the identified issues.

Figure 13: The prompt used during the iterative refinement loop. It provides the *Generator* with the *Critic*'s feedback to guide the revision process.

> **Prompt for Quality Assurance *Judge* Agent**
>
> **Your Task:** You are a QA Verifier for Medical Case Simulations. Perform a rigorous, two-part validation on the provided question set by adopting two distinct personas.
> **Core Directive: Information Segregation**
>
> - **Privileged Information (For Factual Validation Only):** The "Source Case Report" and "Image Captions". This is 100% invisible to the test-taker.
> - **Test-Taker Visible Information (For Logical Solvability Only):** The initial scenario, question stems, images (without captions), and answers from previous steps.
>
> **Your Two-Part Validation Task for Each Question:**
> **1. Factual Validation (as a "Backend Auditor"):**
>
> - Using the **Privileged Information**, is the question's premise, data, and correct answer factually correct and perfectly consistent with the source case?
>
> **2. Logical Solvability (as an "Examinee"):**
>
> - Using **ONLY the Test-Taker Visible Information**, is there a clear, logical path to the single correct answer? Erase all memory of the privileged information for this check.
>
> **Output Format Requirement:** You MUST output your assessment as a JSON object. For each MCQ, provide a "factual_validation" and a "logical_solvability" block, each containing a boolean verdict ("is_factually_correct" / "is_logically_solvable") and a detailed "explanation".

Figure 14: The prompt for the *Judge* agent, which enforces a strict two-part validation focused on factual correctness and logical solvability.

First, as a "backend auditor," it performs *Factual Validation* by comparing the question against the source case. Second, as an "examinee," it performs *Logical Solvability* by assessing if the answer can be deduced using only the information presented up to that point. The agent must output its findings in a structured JSON format, giving a clear verdict for each criterion.

**Instructions for Physician Reviewers**   Questions that pass the AI pre-screening are then subjected to final verification by human clinical experts. The physicians are provided with the same materials and instructed to follow the exact same two-part validation protocol as the *Judge* agent. They first act as auditors, using the source case to perform ***Factual Validation***. Then, they switch to an examinee's perspective to confirm ***Logical Solvability***. This parallel process ensures our principle of conservatism is upheld: any question identified with a potential flaw by either the AI or the human experts is rigorously scrutinized and ultimately rejected if the issue cannot be resolved.

## H   TRANSLATION QUALITY VALIDATION

To facilitate the rigorous expert review process with our native Chinese-speaking physicians, all English-language question sets were systematically translated. Ensuring the fidelity of this translation was paramount. To this end, we conducted a comparative validation study to quantify the consistency

---

**Prompt for Comparative Translation Validation**

**Goal:** Your task is to determine if two Chinese translations of the same medical content are clinically and semantically identical.

**Context:** You will be presented with pairs of text segments (e.g., a question stem, an option, or a scenario).

- **Translation A:** The text produced by our AI system.

- **Translation B:** The text produced by a professional human medical translator (this is your reference).

**CRITICAL:** You will NOT see the original English source. Your judgment must be based solely on comparing Translation A and Translation B.

**Core Evaluation Question:** Would a medical professional reading either translation arrive at the exact same clinical understanding? Consider all aspects:

1. **Factual Equivalence:** Are all medical facts, numbers, units, and terminologies identical?

2. **Nuance Equivalence:** Is the degree of certainty, tone, and clinical implication the same in both versions (e.g., "可能" vs. "确诊")?

3. **Logical Equivalence:** Does the logic of the question and the relationship between premise and conclusion remain unchanged?

**Your Task: Provide a Verdict for Each Pair** For each pair of translations, please provide one of the following two verdicts:

**Verdict: Consistent** Choose this if Translation A is clinically and semantically identical to the reference Translation B. A medical professional would interpret both in exactly the same way.

**Verdict: Inconsistent** Choose this if there is any meaningful difference between the two translations that could potentially alter clinical understanding, even if it's a subtle change in nuance or terminology.

**Final Output:** For each pair, provide your verdict ("Consistent" or "Inconsistent"). If you choose "Inconsistent," please briefly describe the discrepancy.

---

Figure 15: The prompt provided to native Chinese-speaking physicians to validate the consistency between AI-generated and human-expert translations.

between translations generated by our *o3*-powered API and those produced by professional human translators.

**Validation Methodology and Results** We randomly sampled 100 complete question sets for this study. For each set, we generated two parallel Chinese versions: the first was produced by our *o3*-powered API, while the second, serving as a gold-standard reference, was created by two professional biomedical translators. These two translated versions were then presented side-by-side to a panel of three native Chinese-speaking physicians for a comparative review.

Crucially, the physicians did not see the original English text. Their task was to determine if the two Chinese versions were clinically and semantically identical. A pair was marked 'consistent' only if the *o3* translation conveyed the exact same medical facts, nuances, and logical relationships as the expert human translation. The final inter-translation consistency rate was **99.2%**, confirming that the API-powered translation is a highly reliable substitute for manual expert translation for this task.

**Instructions for Comparative Review** To standardize the assessment, each physician reviewer was guided by the prompt detailed in Figure 15. The prompt instructed them to perform a direct comparison and provide a binary judgment on the semantic equivalence of the two translations.

# I SUPPLEMENTARY DETAILS OF PHYSICIAN SCREENING

## I.1 INTER-PHYSICIAN AGREEMENT STATISTICS

Verification comprises an *Annotation* phase, in which each question is labeled by a single physician, followed by an *Inspection* phase conducted by another physician. Any discrepancy triggers a revision loop with the original annotator until consensus is reached, and disagreement statistics are computed using the number of loops as a proxy for inter-annotator agreement. As shown in Table 5,

| | Loop 1 | Loop 2 | Loop 3 |
|---|---|---|---|
| **Case Num** | 1286 (91.4%) | 109 (7.8%) | 12 (0.9%) |

Table 5: Case counts and proportions for each revision loop in physician verification.

| Major Specialty (Total, %) | Subspecialty | Count |
|---|---|---|
| Internal Medicine (95, 39.8%) | Cardiology | 8 |
| | Nephrology | 4 |
| | Gastroenterology | 7 |
| | Respiratory Medicine | 5 |
| | Hematology | 5 |
| | Endocrinology | 3 |
| | General Internal Medicine / Geriatrics | 63 |
| Surgery (72, 30.1%) | General Surgery | 38 |
| | Thoracic Surgery | 7 |
| | Hepatobiliary Surgery | 10 |
| | Urology | 13 |
| | Orthopedics | 5 |
| | Neurosurgery | 4 |
| | Burn Surgery | 2 |
| | Plastic / Cosmetic Surgery | 3 |
| Pediatrics (27, 11.3%) | General Pediatrics | 19 |
| | Pediatric Oncology | 3 |
| | Pediatric Cardiology / Neurology | 5 |
| Obstetrics & Gynecology (21, 8.8%) | Obstetrics | 13 |
| | Gynecology | 8 |
| Dermatology (19, 7.9%) | Dermatology | 19 |
| Neurology (9, 3.8%) | Neurology | 9 |
| Ophthalmology (9, 3.8%) | Ophthalmology | 9 |
| Emergency Medicine (6, 2.5%) | Emergency Medicine | 6 |
| Rehabilitation Medicine (6, 2.5%) | Rehabilitation Medicine | 6 |
| Medical Imaging / Radiology (6, 2.5%) | Diagnostic Radiology | 4 |
| | Medical Imaging (incl. Interventional) | 2 |
| Traditional Chinese Medicine (6, 2.5%) | TCM Internal Medicine | 6 |
| Other Specialties (3, 1.3%) | Oncology | 3 |
| Pathology (1, 0.4%) | Pathology | 1 |

Table 6: Specialty distribution of the 239 physicians in the verification cohort.

disagreements occurred in 8.7% of reviewed cases, with 91.4% resolved after the first loop, 7.8% after the second, and only 0.9% requiring a third. This distribution demonstrates high agreement among physicians and confirms that the two-phase process ensures consistency while keeping revision overhead minimal.

## I.2 CLINICAL SPECIALTY DISTRIBUTION

The physician review cohort consists of 239 licensed physicians spanning all major clinical specialties and subspecialties. Each case is assigned to reviewers with expertise matching the disease category, ensuring domain-appropriate evaluation. This comprehensive coverage and targeted assignment minimize the risk of any disease type lacking specialized review. As shown in Table 6, Internal

> **Clustering Logic for Clinical Stages**
>
> **Principle:** Map granular, AI-generated stage labels to broader categories based on clinical keywords.
> **Example Mappings:**
>
> - Keywords like "initial", "presentation", "assessment" → **Presentation & Assessment**
> - Keywords like "imaging", "diagnostic", "pathology", "histology" → **Diagnosis & Interpretation**
> - Keywords like "management", "therapy", "surgical", "planning" → **Therapeutic Strategy**
> - Keywords like "complication", "deterioration", "adverse" → **Complication Management**
> - Keywords like "follow-up", "surveillance", "post-operative" → **Follow-up**

Figure 16: The expert-defined logic used to cluster granular, AI-generated stage labels into the five primary clinical workflow categories.

Medicine comprises the largest share of the team (39.8%), dominated by General Internal Medicine and Geriatrics. Surgery accounts for 30.1%, with notable representation from General Surgery, Urology, and Hepatobiliary Surgery. Pediatrics, Obstetrics & Gynecology, Dermatology, and Neurology further broaden patient coverage, while smaller yet essential fields, including Emergency Medicine, Rehabilitation, Radiology, Oncology, and Pathology, are also represented.

## J ANNOTATION OF CLINICAL STAGES

To analyze the sequential nature of clinical reasoning, we categorized each question into one of five primary workflow stages. This was achieved through a detailed, two-step clustering process designed to systematically organize the diverse stage labels produced by the AI into a consistent, analyzable framework.

**Initial AI Labeling** During the examination generation stage (Stage 2), our *Generator* agent assigned a descriptive, free-text "stage" label to each question. This initial step resulted in hundreds of unique, granular labels that captured the specific context of each question, such as "Post-operative follow-up," "Initial diagnostic imaging," or "Surgical Intervention planning."

**Expert-Guided Clustering** While granular, these free-text labels were too diverse for a high-level statistical analysis. We, therefore, implemented a two-level clustering protocol under physician supervision. First, we programmatically grouped the AI-generated labels into 8 intermediate clinical categories based on a set of keywords (e.g., all labels containing "therapy" or "surgical" were grouped together). Following this, our physician team reviewed and further consolidated these intermediate groups into the five final, clinically coherent workflow phases shown in Figure 5A. For example, the intermediate categories for "Therapeutic Planning" and "Surgical Intervention" were both mapped to the final *Therapeutic Strategy* stage. This expert-guided logic, which underpins the entire categorization, is summarized in Figure 16.

## K ANNOTATION OF MULTIMODAL CONTENT

The classification of all images and tables into standardized types followed a three-step, AI-human hybrid protocol. This approach was designed to leverage the scalability of AI while ensuring the uncompromising accuracy required for a medical benchmark.

1. **AI Classification:** We first employed the *o3* model to perform an initial classification of all **3,757 images** and **634 tables**. Guided by the prompt in Figure 17, the model assigned a specific, descriptive label (e.g., "CT Scan of the Abdomen," "Table of Baseline Patient Demographics") to each item based on its content and original caption.

2. **Physician Validation:** The complete set of AI-generated labels was then meticulously reviewed by our physician team. This crucial step served as a 100% audit of the AI's output. The experts verified the correctness of the AI's classification with **100% accuracy**, confirming its reliability for this task.

---

**Prompt for Initial Modality Classification**

**Your Task:** You are a medical data specialist. Your task is to classify the given item (image or table) into its most specific and accurate type.
**Input:** You will be given the item's original caption and its content (for tables) or the image itself.
**Instruction:** Provide a concise, specific label describing the item.

- **For Images:** e.g., "CT Scan of the Abdomen," "Chest X-ray (PA view)," "H&E Stained Pathological Slide."

- **For Tables:** e.g., "Table of Baseline Patient Demographics," "Table of Serial Blood Test Results."

---

Figure 17: The prompt used to instruct the *o3* model to perform initial classification of images and tables.

---

**Clustering Logic for Modalities**

**Principle:** Map specific, validated labels into broader analytical categories based on expert-defined rules.
**Image Clustering Examples:**

- Labels like "CT Angiography", "Helical Tomography" → **CT Scans**

- Labels like "Radiography", "Fluoroscopy" → **Radiography & X-ray**

- Labels like "Fundus Photograph", "Dermoscopy" → **Clinical & Surface Photography**

**Table Clustering Examples:**

- Labels like "Blood Test Results", "Biochemistry Panel" → **Lab Results**

- Labels like "Patient Demographics", "Vital Signs Table" → **Patient Demographics & Characteristics**

- Labels like "Longitudinal Treatment Response" → **Treatment Outcomes & Monitoring**

---

Figure 18: The expert-defined logic used to group specific, validated modality labels into the broader categories shown in the final statistics.

3. **Expert-Guided Clustering:** Finally, to create clear, high-level categories for analysis, these specific and now-validated labels were grouped based on expert-defined rules, as summarized in Figure 18. This consolidation, conducted under physician supervision, allowed us to analyze broad trends. For example, validated labels like "CT Angiography" and "Helical Tomography" were grouped into the "CT Scans" category.

## L  STAGE-SPECIFIC DISTRIBUTION OF MULTIMODAL QUESTIONS

In the dataset of 1,407 cases, all contain multimodal information, with modality composition shown in Figures 5C and 5D. To assess variation in multimodal demands along the clinical pathway, we calculated the proportion of questions requiring multimodal interpretation at different progression stages. As shown in Table 7, such items are most frequent in the initial 20%, decline steadily from 20% to 80%, and rise again in the final 20%. This trend suggests heavier multimodal cognitive demands during early diagnostic assessment, reduced use in mid-path management, and greater integration of diverse data sources in final decision and outcome stages.

## M  EVALUATION PROTOCOL DETAILS AND ROBUSTNESS ANALYSES

### M.1  EVALUATION PROTOCOL DETAILS

This section provides a detailed description of the models, parameters, and procedures used in our evaluation.

**Model Suite and Parameters** We evaluated a comprehensive suite of 24 models, which can be grouped into three main categories:

- **Proprietary Models (13):** Leading closed-source models from major AI labs.

| Clinical Pathway Stage | 0–20% | 20–40% | 40–60% | 60–80% | 80–100% |
|---|---|---|---|---|---|
| Proportion | 62.0% | 46.0% | 35.5% | 27.5% | 38.5% |

Table 7: Proportion of multimodal questions across clinical pathway stages, representing the percentage of questions requiring multimodal interpretation within each stage.

---

**Prompt for Evaluation**

**Scenario:**
Initial clinical scenario text, along with any associated figures and tables.
**Previous Q&A Turns:**
Full text of previous questions, options, and the model's prior answers are inserted here for context.
**Current Question:**
Text of the current question.
**Options:**
A. Option A text B. Option B text ... J. Option J text
Please provide the letter of the correct option, formatted as \boxed{LETTER} (e.g., \boxed{A}).

---

Figure 19: The conversational prompt structure used for evaluation. Context from previous turns is cumulatively added for subsequent questions within the same case.

The models tested were: *o3*, *gpt-5*, *gemini-2.5-pro*, *gpt-4.1*, *claude-3-5-sonnet-20241022*, *claude-3-7-sonnet-20250219-thinking*, *gemini-1.5-pro*, *claude-3-5-haiku-20241022*, and their high-efficiency variants: *gpt-4.1-mini-2025-04-14*, *gemini-2.5-flash*, *gemini-2.0-flash*, *gpt-4o-2024-11-20*, and *gpt-4o-mini-2024-07-18*.

- **Open-Source General-Purpose VLMs (7):** Powerful, publicly available Vision-Language Models.

  The models tested were: *Qwen2.5-VL-72B-Instruct*, *Qwen2.5-VL-32B*, *Qwen2.5-VL-7B-Instruct* (Bai et al., 2025), *InternVL3.5-38B*, *InternVL3.5-241B-A28B*, *InternVL3.5-8B* (Wang et al., 2025), *GLM-4.1V-9B* (Team et al., 2025c), and *MiMo-VL-7B-RL-2508* (Team et al., 2025a).

- **Open-Source Medical-Specific VLMs (4):** Models specifically fine-tuned on medical data.

  The models tested were: *medgemma-27b-it*, *medgemma-4b-it* (Sellergren et al., 2025), *HuatuoGPT-Vision-7B* (Chen et al., 2024), *LingShu-7B* and *LingShu-32B* (Team et al., 2025b).

To balance reproducibility with optimal performance, *temperature* was set to 0 for most models. For those featuring specific reasoning modes (e.g., "thinking" variants), we adopted their official recommended inference configurations. If an API call failed due to transient errors like rate limits or timeouts, it was automatically retried up to two times with a brief delay.

**Prompting Strategy** Our protocol maintains a continuous conversation history for each case. The prompt for each question is structured to build upon all previous turns. Figure 19 illustrates the general format. For the first question in a case, only the *Scenario* and *Current Question* blocks are used. For subsequent questions, the *Previous Q&A Turns* block is populated with the full history of preceding questions and the model's own answers. This conversational context is critical for testing sequential reasoning.

**Answer Parsing** To robustly extract the model's final choice, we implemented a hierarchical parsing strategy. The system prioritizes answers enclosed in a *\boxed{}* command (e.g., *\boxed{A}*). If this is not found, it searches for explicit keywords (e.g., "The correct answer is: A"). As a final fallback, it looks for single-letter answers at the end of the response. This multi-layered approach ensures high-fidelity extraction of the model's intended answer.

### M.2 PHYSICIAN BASELINE EVALUATION

The baseline cohort comprised 78 licensed physicians, including 43 Attending Physicians, accounting for 55.1% of the cohort, and 35 senior physicians, of whom 19 were Associate Chief Physicians and 16 were Chief Physicians, accounting together for 44.9%. One hundred patient cases were randomly sampled from the LiveClin dataset, each containing multiple sequential questions spanning diagnostic and management stages. Cases were assigned to physicians whose clinical specialty matched the

| Major Specialty | Subspecialty | Count |
|---|---|---|
| Internal Medicine (31) | Cardiology | 3 |
| | Nephrology | 1 |
| | Gastroenterology | 2 |
| | Respiratory Medicine | 2 |
| | Hematology | 2 |
| | Endocrinology | 1 |
| | General Internal Medicine / Geriatrics | 20 |
| Surgery (23) | General Surgery | 12 |
| | Thoracic Surgery | 2 |
| | Hepatobiliary Surgery | 3 |
| | Urology | 4 |
| | Orthopedics | 2 |
| | Neurosurgery | 1 |
| Pediatrics (9) | General Pediatrics | 6 |
| | Pediatric Oncology | 1 |
| | Pediatric Cardiology / Neurology | 2 |
| Obstetrics & Gynecology (7) | Obstetrics | 4 |
| | Gynecology | 3 |
| Dermatology (6) | Dermatology | 6 |
| Neurology (3) | Neurology | 3 |
| Ophthalmology (3) | Ophthalmology | 3 |
| Emergency Medicine (2) | Emergency Medicine | 2 |
| Rehabilitation Medicine (2) | Rehabilitation Medicine | 2 |
| Medical Imaging / Radiology (2) | Diagnostic Radiology | 1 |
| | Medical Imaging (incl. Interventional) | 1 |
| Traditional Chinese Medicine (2) | TCM Internal Medicine | 2 |

Table 8: Specialty distribution of physicians in the baseline evaluation cohort, proportionally scaled from the full verification team to 78 participants.

disease category to ensure domain-appropriate evaluation. For each case, six independent responses were collected: three from Attending Physicians and three from Associate Chief or Chief Physicians. Accuracy for each group was calculated as the proportion of cases in which all sequential questions were answered correctly. The clinical specialty composition of the cohort is presented in Table 8.

## M.3 EVALUATION ROBUSTNESS ANALYSIS

To assess the robustness of our evaluation paradigm and verify that observed performance differences were not artifacts of prompt design or random variability, we conducted two complementary experiments: repeated benchmark runs under identical settings, and few-shot prompting trials.

**Repeated Runs for Stability** Each model was evaluated in three independent benchmark runs under identical conditions, with `temperature` fixed to 0 for most models. For models that provide proprietary reasoning modes, we followed the officially recommended inference settings. Performance in each run was measured by Case Accuracy, and run to run variability was quantified using the standard deviation across the three runs. As reported in Table 9, most models yielded identical or near identical results, with standard deviations at or below the reporting resolution. Reasoning oriented models exhibited slightly higher variability up to 0.45 percentage points, while several closed source models evaluated with non zero `temperature` showed minor fluctuations

| Model | Run 1 (%) | Run 2 (%) | Run 3 (%) | Std. Dev. |
|---|---|---|---|---|
| GPT-5 | 35.5 | 35.4 | 35.6 | 0.22 |
| o3 | 35.7 | 35.2 | 35.9 | 0.36 |
| GPT-4.1 | 31.8 | 31.7 | 31.9 | 0.10 |
| GPT-4.1-mini | 26.1 | 26.0 | 26.1 | 0.00 |
| GPT-4o | 14.7 | 14.7 | 14.8 | 0.12 |
| GPT-4o-mini | 14.8 | 14.8 | 14.9 | 0.08 |
| InternVL 3.5-241B | 34.4 | 34.4 | 34.3 | 0.00 |
| InternVL 3.5-38B | 18.0 | 18.0 | 18.1 | 0.00 |
| InternVL 3.5-8B | 12.4 | 12.4 | 12.5 | 0.00 |
| Gemini-2.5 Pro | 33.7 | 33.2 | 33.9 | 0.35 |
| Gemini-2.5 Flash | 19.0 | 19.0 | 19.1 | 0.08 |
| Gemini-2.0 Flash | 21.4 | 21.4 | 21.3 | 0.08 |
| Gemini-1.5 Pro | 17.7 | 17.7 | 17.6 | 0.08 |
| Claude-3.7 Sonnet | 25.6 | 25.2 | 25.8 | 0.30 |
| Claude-3.5 Sonnet | 27.3 | 27.0 | 27.5 | 0.25 |
| Claude-3.5 Haiku | 15.3 | 15.3 | 15.2 | 0.08 |
| LingShu-32B | 20.1 | 20.1 | 20.0 | 0.00 |
| LingShu-7B | 15.3 | 15.3 | 15.4 | 0.00 |
| GLM-4.1V-9B | 18.8 | 18.8 | 18.9 | 0.00 |
| MiMo-VL-7B-RL-2508 | 10.9 | 10.6 | 11.0 | 0.21 |
| Qwen2.5-VL-72B | 18.6 | 18.6 | 18.7 | 0.12 |
| Qwen2.5-VL-32B | 17.2 | 17.2 | 17.3 | 0.12 |
| Qwen2.5-VL-7B | 7.9 | 7.9 | 8.0 | 0.12 |
| MedGemma-27B It | 15.6 | 15.6 | 15.5 | 0.00 |
| MedGemma-4B It | 7.5 | 7.5 | 7.6 | 0.00 |
| HuatuoGPT-Vision-7B | 9.2 | 9.2 | 9.3 | 0.00 |

Table 9: Benchmark stability across three independent runs. Scores are reported as Case Accuracy (%), with standard deviations across runs.

from 0.08% to 0.12%. These variations did not affect the relative performance ranking, indicating that the evaluation results are stable and replicable.

**Few-Shot Prompting Trials** To assess whether the zero shot conversational format disadvantages particular models, we reran the benchmark for all models under zero shot, one shot, and three shot prompting conditions. In the one shot and three shot settings, exemplars from the same domain as the target question were placed before evaluation while preserving the conversational prompt structure described in Appendix M. As shown in Table 10, most models changed in Case Accuracy by no more than 0.8%, suggesting that well optimized instruction following models do not rely on few shot adaptation to perform effectively on this benchmark. Several open source medical specific VLMs, including MedGemma-27B, LingShu-32B, and HuatuoGPT-V, showed noticeable declines under few shot prompting. These decreases are likely due to the prompt length exceeding the context length that was emphasized during training.

**Interpretation** Overall, the results demonstrate that the conversational zero-shot evaluation protocol, combined with fixed or officially recommended temperature settings, provides a stable and unbiased foundation for model comparison. The minimal variability across repeated runs, together with the small effect of few-shot prompting for most models, indicates that observed performance differences primarily reflect real disparities in sequential clinical reasoning capability rather than artifacts stemming from prompt configuration or stochastic output variance.

| Model | Zero-shot (%) | One-shot (%) | Three-shot (%) |
|---|---|---|---|
| GPT-5 | 35.5 | 35.8 (+0.3) | 35.1 (-0.4) |
| GPT-4.1 | 31.8 | 31.5 (-0.3) | 32.3 (+0.5) |
| InternVL 3.5-24B | 34.4 | 34.7 (+0.3) | 34.9 (+0.5) |
| Gemini-2.5 Pro | 33.7 | 33.1 (-0.6) | 32.9 (-0.8) |
| Claude-3.5 Sonnet | 27.3 | 27.0 (-0.3) | 27.9 (+0.6) |
| Qwen2.5-VL-72B | 17.2 | 17.4 (+0.2) | 17.0 (-0.2) |
| MedGemma-27B | 15.6 | 14.6 (-1.0) | 13.5 (-2.1) |
| LingShu-32B | 20.1 | 18.4 (-1.7) | 18.0 (-2.1) |
| HuatuoGPT-V-7B | 9.2 | 8.0 (-1.2) | 5.7 (-3.5) |

Table 10: Case Accuracy (%) under different shot settings. Values in parentheses denote changes relative to zero-shot performance.

---

**Prompt for Physician Attribution**

**Task:** Determine the principal factor responsible for any increase in question difficulty between the original and modified sets.

**Steps:**

1. Review the original and modified question sets in parallel.

2. Summarize the specific modifications that contributed to the observed difficulty change.

**Constraint:** Base all judgments strictly on the information contained in the question text and associated clinical context; avoid inference beyond the presented data.

---

Figure 20: Instruction prompt for physicians during attribution analysis.

## N  DETAILS ABOUT ATTRIBUTION ANALYSIS

This analysis was undertaken to determine whether the increased difficulty observed in the ablation study is attributable to genuine clinical complexity rather than artifacts introduced during question generation. The study involved 20 licensed physicians, each with a minimum of five years of post-certification clinical practice. Collectively, the cohort represented major clinical domains including internal medicine, surgery, pediatrics, emergency medicine, and radiology. Review assignments were matched to each physician's domain expertise to ensure content-appropriate evaluation.

**Procedure** The attribution process followed the same two phase protocol used in the main benchmark quality assurance workflow. In the Annotation phase, an attending physician reviewed paired sets of questions from the original configuration and from the modified set produced by a workflow change, then identified the primary factor that caused a shift from trivial to non trivial classification. In the Inspection phase, a senior physician independently reviewed each attribution. When disagreement occurred, the original annotator revised the attribution and the case was re examined until agreement was reached. Only attributions that achieved consensus in the second phase were retained for analysis. The instruction prompt given to participating physicians is shown in Figure 20.

**Results** Consensus analysis classified transitions into five categories across two comparison groups. In the comparison where Physician authorship was replaced by the Generator with $n = 44$, the main factors were retention of authentic but incomplete or noisy clinical information in 22 cases, which introduced decision making under uncertainty. Another factor was the incorporation of cross stage reasoning that spans multiple phases of the clinical pathway in 16 cases. A third factor was the addition of clinically similar differential diagnoses in 9 cases, which increased the subtlety required for discriminative reasoning.

In the comparison where the Critic was introduced into the Generator workflow with $n = 23$, the main factors were an expansion of multi step reasoning demands, such as requiring extraction of intermediate features before selecting a treatment, which occurred in 16 cases. Another factor was refinement of distractor plausibility in 10 cases, where implausible alternatives were replaced with

> **Prompt for Physician Rarity Annotation**
>
> **Task:** Determine whether the condition described in the case report is rare or unrare.
> **Steps:**
>
> 1. Review the case report content and underlying diagnosis.
>
> 2. Apply domain knowledge and available prevalence data to decide on rarity.
>
> 3. If prevalence data are inconclusive, rely on clinical experience in general practice.
>
> **Constraint:** Base all judgments strictly on the information in the case report and widely accepted medical knowledge; avoid speculative assumptions.

Figure 21: Instruction prompt for physicians during rarity annotation.

| Loop Num | 1 | 2 | 3 |
|---|---|---|---|
| **Case Number (Proportion)** | 1276 (90.7%) | 113 (8.0%) | 18 (1.1%) |

Table 11: Inter-annotator agreement across annotation loops for rarity classification. Case counts are shown with their proportion in parentheses.

clinically credible but subtly incorrect options. Across all transitions, more than 80% were attributed to modifications that reflect genuine clinical reasoning challenges rather than artifactual changes introduced by the generation process.

**Interpretation** The findings support the conclusion that the observed increases in difficulty were predominantly the result of intentional augmentation of realistic clinical problem features, such as the preservation of noisy or incomplete data, the structuring of questions to require integration across multiple clinical stages, and the elevation of distractor quality to match plausible differential diagnoses. These changes align with the benchmark's aim of simulating authentic and challenging decision-making scenarios.

## O   PRIVATE LEADERBOARD FOR CONTAMINATION MONITORING

To mitigate the risk of benchmark exploitation from rapid model iterations by individual developers, we operate a smaller private leaderboard updated monthly. This setup enables timely detection of potential contamination events before they can affect the public benchmark.

**Construction** Two evaluation sets of 140 items each were constructed from case reports published in July and August 2025, following the same physician–AI collaborative workflow used for the main benchmark. To achieve rapid turnaround for monthly monitoring, we sampled three cases per 72 Level-2 disease clusters, compared with thirty per category in the biannual public release. All items underwent the same clinician validation to ensure quality parity with the public benchmark.

**Evaluation Protocol** The private leaderboard employs identical prompting format, inference parameters, and scoring methodology as described for the main benchmark in Appendix M.

## P   RARITY ANNOTATION AND IMPACT ANALYSIS

This analysis assesses the potential bias arising from the overrepresentation of rare conditions in PubMed case reports.

**Annotation** Because there is no universally accepted prevalence based definition of rare that applies to case reports across multiple specialties, we used a standardized protocol in which licensed physicians labeled each case as rare or unrare based on clinical experience and judgment. Annotation was performed by one domain specialist physician and independently reviewed by another physician. Any disagreements were resolved through discussion until consensus was reached. This procedure achieved high first round inter annotator agreement of 90.7% as shown in Table 11. The instructions provided to annotators are shown in Figure 21.

| Model | Rare (%) | Unrare (%) | Δ | Overall (%) |
|---|---|---|---|---|
| GPT-5 | 35.65 | 35.05 | -0.60 | 35.5 |
| O3 | 35.99 | 34.11 | -1.88 | 35.7 |
| GPT-4.1 | 31.92 | 30.84 | -1.08 | 31.8 |
| GPT-4.1-mini | 26.32 | 25.00 | -1.32 | 26.1 |
| GPT-4o | 14.23 | 17.76 | 3.53 | 14.7 |
| GPT-4o-mini | 14.15 | 19.34 | 5.19 | 14.8 |
| InternVL 3.5-241B | 33.89 | 37.12 | 3.23 | 34.4 |
| InternVL 3.5-38B | 17.28 | 21.78 | 4.50 | 18.0 |
| InternVL 3.5-8B | 11.63 | 16.48 | 4.85 | 12.4 |
| Gemini-2.5 Pro | 33.36 | 35.05 | 1.69 | 33.7 |
| Gemini-2.5 Flash | 18.54 | 20.56 | 2.02 | 19.0 |
| Gemini-2.0 Flash | 20.61 | 25.00 | 4.39 | 21.4 |
| Gemini-1.5 Pro | 17.02 | 20.28 | 3.26 | 17.7 |
| Claude-3.7 Sonnet | 25.06 | 29.91 | 4.85 | 25.6 |
| Claude-3.5 Sonnet | 27.01 | 29.44 | 2.43 | 27.3 |
| Claude-3.5 Haiku | 14.92 | 19.63 | 4.71 | 15.3 |
| LingShu-32B | 20.00 | 20.40 | 0.40 | 20.1 |
| LingShu-7B | 15.24 | 15.64 | 0.40 | 15.3 |
| GLM-4.1V-9B | 17.56 | 21.96 | 4.40 | 18.8 |
| MiMo-VL-7B-RL-2508 | 10.32 | 14.49 | 4.17 | 10.9 |
| Qwen2.5-VL-72B | 17.87 | 21.96 | 4.09 | 18.6 |
| Qwen2.5-VL-32B | 16.49 | 21.50 | 5.01 | 17.2 |
| Qwen2.5-VL-7B | 7.73 | 8.41 | 0.68 | 7.9 |
| MedGemma-27B It | 14.76 | 20.04 | 5.28 | 15.6 |
| MedGemma-4B It | 7.56 | 7.48 | -0.08 | 7.5 |
| HuatuoGPT-Vision-7B | 8.73 | 13.08 | 4.35 | 9.2 |

Table 12: Model Case Accuracy (%) on rare (1181 cases, 84%) and unrare (226 cases, 16%) subsets. Δ denotes the difference between unrare and rare performance.

**Nature of Unrare Cases in the Dataset** In addition to reports on rare diseases, PubMed case reports also include a substantial number of cases that are clinically unrare. These cases often involve conditions commonly encountered in routine practice but are reported because of distinctive features such as unusual symptom constellations, unexpected treatment responses, novel mechanistic insights, or innovative clinical interventions. In many reports, the focus is on atypical aspects of diagnosis, management, or pathophysiology rather than on low disease prevalence. Accordingly, the unrare category in our dataset comprises common or moderately common conditions presented in distinctive clinical contexts that offer educational or research value.

**Evaluation and Findings** An unrare subset derived from the annotated set was used for comparative evaluation. We measured model performance using the same Case Accuracy metric and the same prompting and scoring protocols as in the main benchmark. As shown in Table 12, most models exhibited absolute accuracy differences below 5% between the unrare subset and the full set. Higher performing models were largely insensitive to rarity, while some GPT variants performed marginally better on rare cases. These small differences and stable rankings suggest that rarity related bias has negligible impact on evaluation outcomes, supporting the use of PubMed case reports for benchmark construction within our intended scope despite their relatively high proportion of rare conditions.

