- https://doi.org/10.1155/crdm/1258930

- https://doi.org/10.3389/fnins.2025.1593659

- https://doi.org/10.7759/cureus.79254

- https://doi.org/10.26603/001c.137951

- https://doi.org/10.35371/aoem.2025.37.e7

- https://doi.org/10.1177/25424823251342487

- https://doi.org/10.1186/s13256-025-05180-8

- https://doi.org/10.7759/cureus.79818

- https://doi.org/10.1155/crit/2822007

- https://doi.org/10.4330/wjc.v17.i2.101588

- https://doi.org/10.7759/cureus.78952

- https://doi.org/10.7759/cureus.84310

- https://doi.org/10.4103/jpbs.jpbs_125_25

- https://doi.org/10.1016/j.ijscr.2025.111027

- https://doi.org/10.7759/cureus.84465

- https://doi.org/10.4240/wjgs.v17.i3.100951

- https://doi.org/10.1177/2050313X251321046

- https://doi.org/10.7759/cureus.78110

- https://doi.org/10.7759/cureus.78403

- https://doi.org/10.1016/j.inpm.2025.100545

- https://doi.org/10.3389/fimmu.2025.1527102

- https://doi.org/10.7762/cnr.2025.14.1.1

- https://doi.org/10.1590/1980-5764-DN-2025-0295

- https://doi.org/10.7759/cureus.80788

- https://doi.org/10.7759/cureus.85421

- https://doi.org/10.7759/cureus.85912

- https://doi.org/10.1186/s13256-025-05268-1

- https://doi.org/10.2147/IMCRJ.S510696

- https://doi.org/10.3390/children12040516

- https://doi.org/10.3389/fphys.2025.1609975

- https://doi.org/10.7759/cureus.78099

- https://doi.org/10.3389/fped.2024.1500373

- https://doi.org/10.1097/MD.0000000000041260

- https://doi.org/10.3389/fnins.2025.1570726
- https://doi.org/10.7759/cureus.76800
- https://doi.org/10.1016/j.jpra.2025.02.019
- https://doi.org/10.1016/j.jpra.2025.03.024
- https://doi.org/10.7759/cureus.81080
- https://doi.org/10.7759/cureus.78363
- https://doi.org/10.1002/ccr3.70187
- https://doi.org/10.3389/fcvm.2025.1471462
- https://doi.org/10.7759/cureus.86099
- https://doi.org/10.13107/jocr.2025.v15.i01.5114
- https://doi.org/10.7759/cureus.80501
- https://doi.org/10.7759/cureus.86760
- https://doi.org/10.12659/AJCR.947160
- https://doi.org/10.1002/ccr3.70193
- https://doi.org/10.3390/healthcare13080940
- https://doi.org/10.7759/cureus.86610
- https://doi.org/10.7759/cureus.86898
- https://doi.org/10.7759/cureus.77183
- https://doi.org/10.1002/jgh3.70177
- https://doi.org/10.7759/cureus.79904
- https://doi.org/10.12998/wjcc.v13.i14.102791
- https://doi.org/10.7759/cureus.85258
- https://doi.org/10.2147/JIR.S513138
- https://doi.org/10.7759/cureus.86845
- https://doi.org/10.1093/jscr/rjaf044
- https://doi.org/10.7759/cureus.86176
- https://doi.org/10.3389/fphar.2024.1443609
- https://doi.org/10.7759/cureus.78214
- https://doi.org/10.7759/cureus.81135
- https://doi.org/10.1016/j.toxrep.2025.101968
- https://doi.org/10.7759/cureus.85016
- https://doi.org/10.12998/wjcc.v13.i9.101363
- https://doi.org/10.1186/s13256-025-05330-y
- https://doi.org/10.1002/ccr3.70436

- https://doi.org/10.3389/fimmu.2025.1567377

- https://doi.org/10.70352/scrj.cr.25-0104

- https://doi.org/10.7759/cureus.78166

- https://doi.org/10.7759/cureus.87043

- https://doi.org/10.2176/jns-nmc.2024-0334

- https://doi.org/10.7759/cureus.85232

- https://doi.org/10.21037/tlcr-24-711

- https://doi.org/10.1002/ccr3.70540

- https://doi.org/10.1002/ccr3.70142

- https://doi.org/10.4253/wjge.v17.i1.101119

- https://doi.org/10.7759/cureus.85064

- https://doi.org/10.1016/j.radcr.2024.12.064

- https://doi.org/10.7759/cureus.78999

- https://doi.org/10.7759/cureus.81556

- https://doi.org/10.7759/cureus.80705

- https://doi.org/10.3389/fpsyt.2025.1513022

- https://doi.org/10.7759/cureus.81647

- https://doi.org/10.1002/ccr3.70197

- https://doi.org/10.1002/ccr3.70099

- https://doi.org/10.7759/cureus.85188

- https://doi.org/10.7759/cureus.81186

- https://doi.org/10.1016/j.ijscr.2025.111445

- https://doi.org/10.1016/j.ijscr.2025.111329

- https://doi.org/10.7759/cureus.80043

- https://doi.org/10.1016/j.ijscr.2025.110999

- https://doi.org/10.7759/cureus.77691

- https://doi.org/10.3390/jpm15040155

- https://doi.org/10.1093/ehjcr/ytaf034

- https://doi.org/10.7759/cureus.86113

- https://doi.org/10.7759/cureus.80622

- https://doi.org/10.1097/MD.0000000000041285

- https://doi.org/10.1097/MD.0000000000041166

- https://doi.org/10.1177/2050313X241307682

- https://doi.org/10.7759/cureus.79099

- https://doi.org/10.3934/Neuroscience.2025009

- https://doi.org/10.7759/cureus.83214

- https://doi.org/10.1186/s13256-025-05049-w

- https://doi.org/10.7759/cureus.86992

- https://doi.org/10.7759/cureus.78218

- https://doi.org/10.3389/fimmu.2025.1557565

- https://doi.org/10.1186/s13256-025-05149-7

- https://doi.org/10.1177/03000605251345971

- https://doi.org/10.7759/cureus.83095

- https://doi.org/10.7759/cureus.85673

- https://doi.org/10.7759/cureus.79644

- https://doi.org/10.7759/cureus.77948

- https://doi.org/10.7759/cureus.77582

- https://doi.org/10.1016/j.ijscr.2025.111127

- https://doi.org/10.1097/GOX.0000000000006690

- https://doi.org/10.7759/cureus.78014

- https://doi.org/10.1186/s12886-025-04106-8

- https://doi.org/10.1093/omcr/omae175

- https://doi.org/10.3389/fmed.2025.1609610

- https://doi.org/10.7759/cureus.86568

- https://doi.org/10.1016/j.rmcr.2025.102173

- https://doi.org/10.7759/cureus.79272

- https://doi.org/10.3390/jcm14041104

- https://doi.org/10.7759/cureus.82561

- https://doi.org/10.3389/fimmu.2025.1473190

- https://doi.org/10.7759/cureus.77181

- https://doi.org/10.7759/cureus.79103

- https://doi.org/10.7759/cureus.77435

- https://doi.org/10.3389/fmed.2025.1545428

- https://doi.org/10.7759/cureus.79600

- https://doi.org/10.3389/fsurg.2025.1533174

- https://doi.org/10.7759/cureus.77522

- https://doi.org/10.1515/med-2025-1184

- https://doi.org/10.7759/cureus.77136

- https://doi.org/10.3389/fimmu.2025.1585844
- https://doi.org/10.13107/jocr.2025.v15.i05.5608
- https://doi.org/10.7759/cureus.86225
- https://doi.org/10.7759/cureus.80855
- https://doi.org/10.7759/cureus.81457
- https://doi.org/10.7759/cureus.85425
- https://doi.org/10.7759/cureus.79347
- https://doi.org/10.1186/s12959-025-00752-6
- https://doi.org/10.34172/japid.025.3760
- https://doi.org/10.1002/ccr3.70290
- https://doi.org/10.7759/cureus.83820
- https://doi.org/10.7759/cureus.81247
- https://doi.org/10.7759/cureus.77821
- https://doi.org/10.3389/fphar.2025.1431422
- https://doi.org/10.12659/AJCR.947262
- https://doi.org/10.12659/AJCR.947484
- https://doi.org/10.7759/cureus.79540
- https://doi.org/10.1186/s40981-025-00792-x
- https://doi.org/10.3389/fmed.2025.1564369
- https://doi.org/10.1177/03000605251325655
- https://doi.org/10.7759/cureus.85698
- https://doi.org/10.3892/br.2025.1920
- https://doi.org/10.1016/j.radcr.2025.05.042
- https://doi.org/10.7759/cureus.77979
- https://doi.org/10.51866/cr.782
- https://doi.org/10.1186/s12905-025-03719-x
- https://doi.org/10.7759/cureus.78329
- https://doi.org/10.1002/vms3.70206
- https://doi.org/10.1186/s12871-025-03166-z
- https://doi.org/10.29374/2527-2179.bjvm008924
- https://doi.org/10.3389/fresc.2025.1600145
- https://doi.org/10.7759/cureus.83354
- https://doi.org/10.3389/fmed.2025.1518628
- https://doi.org/10.1002/ccr3.70068

- https://doi.org/10.7759/cureus.78677
- https://doi.org/10.3389/fpsyt.2024.1514153
- https://doi.org/10.7759/cureus.86184
- https://doi.org/10.1002/ccr3.70467
- https://doi.org/10.7759/cureus.80626
- https://doi.org/10.1186/s12871-025-03115-w
- https://doi.org/10.1002/deo2.70114
- https://doi.org/10.4082/kjfm.24.0210
- https://doi.org/10.7759/cureus.78302
- https://doi.org/10.1016/j.rmcr.2025.102201
- https://doi.org/10.7759/cureus.78041
- https://doi.org/10.1002/ccr3.70228
- https://doi.org/10.1016/j.ijscr.2025.111029
- https://doi.org/10.1007/s10006-025-01379-7
- https://doi.org/10.1186/s13256-025-05313-z
- https://doi.org/10.1186/s12883-025-04193-6
- https://doi.org/10.1097/GOX.0000000000006737
- https://doi.org/10.7759/cureus.80433
- https://doi.org/10.7759/cureus.79154
- https://doi.org/10.7759/cureus.80363
- https://doi.org/10.7759/cureus.77801
- https://doi.org/10.7759/cureus.86677
- https://doi.org/10.7759/cureus.82135
- https://doi.org/10.1002/ccr3.70457
- https://doi.org/10.7759/cureus.79269
- https://doi.org/10.1002/pcn5.70091
- https://doi.org/10.3960/jslrt.24080
- https://doi.org/10.62675/2965-2774.20250159
- https://doi.org/10.1002/iju5.70055
- https://doi.org/10.1186/s40780-025-00427-4
- https://doi.org/10.7759/cureus.77387
- https://doi.org/10.1016/j.ijscr.2025.111508
- https://doi.org/10.1016/j.eucr.2025.103100
- https://doi.org/10.1186/s12879-025-10484-7

- https://doi.org/10.3389/fmed.2025.1501242
- https://doi.org/10.7759/cureus.79012
- https://doi.org/10.22037/iej.v20i1.45478
- https://doi.org/10.12998/wjcc.v13.i18.103438
- https://doi.org/10.3390/reports8020042
- https://doi.org/10.7759/cureus.79246
- https://doi.org/10.1016/j.ijscr.2025.110879
- https://doi.org/10.5492/wjccm.v14.i1.97443
- https://doi.org/10.1016/j.radcr.2025.05.075
- https://doi.org/10.1097/MD.0000000000042887
- https://doi.org/10.7759/cureus.84354
- https://doi.org/10.70352/scrj.cr.24-0014
- https://doi.org/10.7759/cureus.77010
- https://doi.org/10.1177/15385744251315998
- https://doi.org/10.1016/j.ijscr.2025.111369
- https://doi.org/10.7759/cureus.79344
- https://doi.org/10.7759/cureus.84939
- https://doi.org/10.1016/j.tcr.2025.101204
- https://doi.org/10.7759/cureus.83301
- https://doi.org/10.1159/000543990
- https://doi.org/10.1159/000544099
- https://doi.org/10.1002/jgf2.70017
- https://doi.org/10.1177/03000605251315353