# OpenReview forum: "LiveClin: A Live Clinical Benchmark without Leakage"
_ICLR.cc/2026/Conference — ICLR 2026 Poster_

### Official Review · Reviewer_KBAZ · 2025-10-25

**Soundness:** 3
**Presentation:** 2
**Contribution:** 2
**Rating:** 4
**Confidence:** 3

**Summary:**

LiveClin tackles the problem of LLM evaluation in healthcare settings. Recognizing the need for benchmarks that are dynamic, to prevent inflation of scores due to contamination, and multi-turn, to evaluate multi-stage clinical reasoning, the authors propose a benchmark they call LiveClin, a novel biannually updated clinical benchmark of over 1,000 multi-question cases. The authors build the benchmark from the last six months of PubMed case reports and evaluate in-depth across LLMs.

LiveClin is generated in a human-AI agentic workflow and constructed to cover comprehensive disease clusters. Evaluation is comprehensive and informative. The authors discover that while individual question accuracy can be high for most models, overall case accuracy is low and decreases with model updates. The detail of the dataset allows for an understanding of model performance by clinical area.

**Strengths:**

-	The benchmark is constructed to be very comprehensive across medical topics. The taxonomy is clear, well explained and well evaluated.
-	The authors include a clear description and depiction of the benchmark generation process and evaluate with creative ablations.

**Weaknesses:**

-	There could be further discussion of the fact that the accuracy for specific questions is high for most models, but that total case accuracy is low. This is an interesting finding of the benchmark consistent with other claims that LLMs lack longitudinal reasoning.
-	It would also be helpful to argue or explain that cases from PubMed will be sufficient for evaluation. What kinds of cases are typically published in this forum? Does this bias the benchmark?
-	There are very few medical specific LLMs tested. This is surprising given the authors claim that domain specific models may be the solution to low overall performance by general models that decreases with newer versions.
-	The authors should compare to other recent medical reasoning benchmarks using case reports including: MedCaseReasoning Wu et al 2025, McDuff et al 2025 (NEJM CPC), and CaseReportBench Zhang et al 2025.

Additional notes (not a part of recommendation)
-	Line 116: Figure is missing an A in NARRATIVE.
-	Line 185: COnstruction
-	Line 266: experts is written twice.
-	Line 273: cut-off in figure could be improved.
-	Table 1: I think it should be “Cost ($).”
-	It is odd that the related work is in the appendix.
-	What is the horizontal line in Figure 7B?

**Questions:**

How will this benchmark be prevented from being contaminated if the data is on open access PubMed? There could be further flushing out of the live generation methods. Will old cases be dropped once they have become available for long enough for models to be trained on? Where will this benchmark be managed and who will have access? How will over 200 physicians be employed regularly to maintain this benchmark?

---

> ### Author Response · Authors · 2025-11-21
> **Reply to KBAZ, Part 1**
>
> Thanks for recognizing the breadth of our benchmark across medical topics, the clarity and rigorous evaluation of our taxonomy, and our detailed depiction of the benchmark generation process with creative ablations. Built on a diverse, clinician‑validated, multimodal dataset, our work aims to provide contamination‑free, and up‑to‑date evaluations. We will address your concerns in detail below.
>
>
>
> > W1: There could be further discussion of the fact that the accuracy for specific questions is high for most models, but that total case accuracy is low. This is an interesting finding of the benchmark consistent with other claims that LLMs lack longitudinal reasoning.
>
> We appreciate this important observation. The discrepancy between high per‑question and low case‑level accuracy is largely explained by compounding error probabilities. For instance, GPT‑5’s per‑question accuracy of 80.4% yields an expected five‑question case accuracy of (0.804)^5 ≈ 33.6% under an independence assumption, close to the empirical 35.5%. Our strict scoring reflects the clinical reality that a single error can have severe consequences. Furthermore, consistent with prior claims that LLMs lack longitudinal reasoning, Figure 7B shows that error rates increase in the later stages of a case, a trend that is more pronounced in open‑source medical models compared with their closed‑source counterparts.

---

> ### Author Response · Authors · 2025-11-21
> **Reply to KBAZ, Part 2**
>
> > W2: It would also be helpful to argue or explain that cases from PubMed will be sufficient for evaluation. What kinds of cases are typically published in this forum? Does this bias the benchmark?
>
> We appreciate your thoughtful comments and share your concern about the representativeness of PubMed cases.
>
> PubMed and PubMed Central are leading repositories of biomedical literature, widely used by clinicians and researchers, and host a diverse range of case reports [1]. And according to [NIH guidelines](https://pmc.ncbi.nlm.nih.gov/articles/PMC5686928/), the case reports are typically published to document unexpected symptom correlations, unforeseen treatment outcomes, new insights into pathogenesis, unique clinical features, or novel therapeutic approaches.
>
> Our benchmark is motivated by the rapid growth of medical knowledge, with a doubling time estimated at 73 days by 2020 [2], making continuously updated evaluations essential. Daily case report publications allow us to capture emerging knowledge in real time, providing coverage that is sufficient for our intended scope and representing a broad range of clinical contexts.
>
> We also recognize that case reports may contain a higher proportion of rare conditions. To assess potential bias, we futher annotate each case as rare or common, and compared model accuracy across these subsets. As shown in the table below, accuracy on common cases differed from overall accuracy by less than 5%, with stronger models less affected by rarity and recent GPT models occasionally performing better on rare cases. These findings suggest that rarity has only a limited impact on performance, ranking, and robustness, and that PubMed case reports remain suitable for evaluation within our benchmark’s scope. Please see lines 514-523 and 1547-1610 in the new version for details.
>
> | Model Name          | Rare (1181, 84%) | Unrare (226, 16%) | Δ(U,R) | OverAll |
> | ------------------- | ---------------- | ----------------- | ------ | ------- |
> | GPT-5               | 35.65%           | 35.05%            | -0.60% | 35.5%   |
> | O3                  | 35.99%           | 34.11%            | -1.88% | 35.7%   |
> | GPT-4.1             | 31.92%           | 30.84%            | -1.08% | 31.8%   |
> | GPT-4.1-mini        | 26.32%           | 25.00%            | -1.32% | 26.1%   |
> | GPT-4o              | 14.23%           | 17.76%            | 3.53%  | 14.7%   |
> | GPT-4o-mini         | 14.15%           | 19.34%            | 5.19%  | 14.8%   |
> | InternVL 3.5-241B   | 33.89%           | 37.12%            | 3.23%  | 34.4%   |
> | nternVL 3.5-38B     | 17.28%           | 21.78%            | 4.50%  | 18.0%   |
> | InternVL 3.5-8B     | 11.63%           | 16.48%            | 4.85%  | 12.4%   |
> | Gemini-2.5 Pro      | 33.36%           | 35.05%            | 1.69%  | 33.7%   |
> | Gemini-2.5 Flash    | 18.54%           | 20.56%            | 2.02%  | 19.0%   |
> | Gemini-2.0 Flash    | 20.61%           | 25.00%            | 4.39%  | 21.4%   |
> | Gemini-1.5 Pro      | 17.02%           | 20.28%            | 3.26%  | 17.7%   |
> | Claude-3.7 Sonnet   | 25.06%           | 29.91%            | 4.85%  | 25.6%   |
> | Claude-3.5 Sonnet   | 27.01%           | 29.44%            | 2.43%  | 27.3%   |
> | Claude-3.5 Haiku    | 14.92%           | 19.63%            | 4.71%  | 15.3%   |
> | LingShu-32B         | 20.00%           | 20.40%            | 0.40%  | 20.1%   |
> | LingShu-7B          | 15.24%           | 15.64%            | 0.40%  | 15.3%   |
> | GLM-4.1V-9B         | 17.56%           | 21.96%            | 4.40%  | 18.8%   |
> | MiMo-VL-7B-RL-2508  | 10.32%           | 14.49%            | 4.17%  | 10.9%   |
> | Qwen2.5-VL-72B      | 17.87%           | 21.96%            | 4.09%  | 18.6%   |
> | Qwen2.5-VL-32B      | 16.49%           | 21.50%            | 5.01%  | 17.2%   |
> | Qwen2.5-VL-7B       | 7.73%            | 8.41%             | 0.68%  | 7.9%    |
> | MedGemma-27B It     | 14.76%           | 20.04%            | 5.28%  | 15.6%   |
> | MedGemma-4B It      | 7.56%            | 7.48%             | -0.08% | 7.5%    |
> | HuatuoGPT-Vision-7B | 8.73%            | 13.08%            | 4.35%  | 9.2%    |
>
> [1]. https://pubmed.ncbi.nlm.nih.gov/about/
>
> [2]. [Crises and Turnaround Management: Lessons Learned from Recovery of New Orleans and Tulane University Following Hurricane Katrina](https://pubmed.ncbi.nlm.nih.gov/30309439/#&gid=article-figures&pid=figure-2-uid-1)

---

> ### Author Response · Authors · 2025-11-21
> **Reply to KBAZ, Part 3**
>
> > W3: There are very few medical specific LLMs tested. This is surprising given the authors claim that domain specific models may be the solution to low overall performance by general models that decreases with newer versions.
>
> We selected these representative models because previous medical evaluations consistently show them achieving the best performance. To present this more intuitively, we report their results below across seven established multimodal clinical benchmarks: OmniMedVQA [1], PMC‑VQA [2], VQA‑RAD [3], SLAKE [4], PathVQA [5], MedXpertQA [6], and MMMU‑Med [7].
>
> We would like to clarify that our intention in the paper is not to suggest that domain‑specific models alone can fully address the performance decline sometimes observed in general models. Rather, we advocate building upon newer and stronger general base models, complemented by targeted domain‑specific optimization, as a promising strategy for the medical AI community to further advance performance.
>
> | Model           | MMMU-Med | VQA-RAD  | SLAKE    | PathVQA  | PMC-VQA | OmniMedVQA | MedXpertQA | Avg.     |
> | --------------- | -------- | -------- | -------- | -------- | ------- | ---------- | ---------- | -------- |
> | BiomedGPT [8]   | 24.9     | 16.6     | 13.6     | 11.3     | 27.6    | 27.9       | -          | -        |
> | Med-R1 [9]      | 34.8     | 39.0     | 54.5     | 15.3     | 47.4    | 5.3        | 21.1       | -        |
> | MedVLM-R1 [10]  | 35.2     | 48.6     | 56.0     | 32.5     | 47.6    | **77.7**   | 20.4       | 45.4     |
> | LLaVA-Med [11]  | 29.3     | 53.7     | 48.0     | 38.0     | 33.5    | 44.3       | 20.3       | 37.8     |
> | MedGemma-4B     | 43.7     | **72.5** | 76.4     | 48.8     | 49.9    | 69.8       | 22.3       | 54.8     |
> | HuatuoGPT-V-7B  | 47.3     | 67.0     | 67.8     | 48.0     | 53.3    | 74.2       | 21.6       | 54.2     |
> | MedGemma-27B-IT | 56.2     | 62.3     | 74.9     | 44.4     | 49.5    | 66.3       | **33.9**   | 55.4     |
> | Lingshu-7B      | 54.0     | 67.9     | 83.1     | 61.9     | 56.3    | 82.9       | 26.7       | 61.8     |
> | Lingshu-32B     | **62.3** | **76.5** | **89.2** | **65.9** | 57.9    | **83.4**   | 30.9       | **66.6** |
>
> [1]. [OmniMedVQA: A New Large-Scale Comprehensive Evaluation Benchmark for Medical LVLM](https://arxiv.org/abs/2402.09181)
>
> [2]. [PMC-VQA: Visual Instruction Tuning for Medical Visual Question Answering](https://arxiv.org/abs/2305.10415)
>
> [3]. [A dataset of clinically generated visual questions and answers about radiology images](https://www.nature.com/articles/sdata2018251)
>
> [4]. [SLAKE: A Semantically-Labeled Knowledge-Enhanced Dataset for Medical Visual Question Answering](https://arxiv.org/abs/2102.09542)
>
> [5]. [PathVQA: 30000+ Questions for Medical Visual Question Answering](https://arxiv.org/abs/2003.10286)
>
> [6]. [MedXpertQA: Benchmarking Expert-Level Medical Reasoning and Understanding](https://arxiv.org/abs/2501.18362)
>
> [7]. [MMMU: A Massive Multi-discipline Multimodal Understanding and Reasoning Benchmark for Expert AGI](https://arxiv.org/abs/2311.16502)
>
> [8]. [BiomedGPT: A Generalist Vision-Language Foundation Model for Diverse Biomedical Tasks](https://arxiv.org/abs/2305.17100)
>
> [9]. [Med-R1: Reinforcement Learning for Generalizable Medical Reasoning in Vision-Language Models](https://arxiv.org/abs/2503.13939)
>
> [10]. [MedVLM-R1: Incentivizing Medical Reasoning Capability of Vision-Language Models (VLMs) via Reinforcement Learning](https://arxiv.org/abs/2502.19634)
>
> [11]. [LLaVA-Med: Training a Large Language-and-Vision Assistant for Biomedicine in One Day](https://arxiv.org/abs/2306.00890)

---

> ### Author Response · Authors · 2025-11-21
> **Reply to KBAZ, Part 4**
>
> > W4: The authors should compare to other recent medical reasoning benchmarks using case reports including: MedCaseReasoning Wu et al 2025, McDuff et al 2025 (NEJM CPC), and CaseReportBench Zhang et al 2025.
>
> We have included in Table below a direct comparison between LiveClin and the mentioned case‑report‑based benchmarks. As shown, LiveClin is the **only** benchmark that simultaneously provides (i) *multimodal clinical evidence*, (ii) *full multi‑stage pathway simulation*, (iii) *dynamic, contamination‑resistant contemporary cases*, and (iv) *will open public evaluation*. Further detailed discussion is provided in line 780–785 of the paper.
>
> | Benchmark            | Multimodal | Multi‑Stage Pathway | Contemporary & Contamination‑Resistant | Public Evaluation |
> | -------------------- | ---------- | ------------------- | -------------------------------------- | ----------------- |
> | MedCaseReasoning [1] | No         | No                  | No                                     | Yes               |
> | McDuff NEJM CPC [2]  | Yes        | Yes                 | No                                     | No (AMIE closed)  |
> | CaseReportBench [3]  | No         | No                  | No                                     | Yes               |
> | LiveClin             | Yes        | Yes                 | Yes                                    | Yes               |
>
> [1]. [MedCaseReasoning: Evaluating and learning diagnostic reasoning from clinical case reports](https://arxiv.org/abs/2505.11733)
>
> [2]. [Towards accurate differential diagnosis with large language models](https://www.nature.com/articles/s41586-025-08869-4)
>
> [3]. [CaseReportBench: An LLM Benchmark Dataset for Dense Information Extraction in Clinical Case Reports](https://arxiv.org/abs/2505.17265)
>
>
>
>
>
> > W additional: Additional notes (not a part of recommendation) - Line 116: Figure is missing an A in NARRATIVE. - Line 185: COnstruction - Line 266: experts is written twice. - Line 273: cut-off in figure could be improved. - Table 1: I think it should be “Cost ($).” - It is odd that the related work is in the appendix. - What is the horizontal line in Figure 7B?
>
> - Typos: All identified typographical errors have been corrected in the latest revision. Figures’ revision is in progress.
> - Related work in the appendix: Due to page‑limit constraints, we follow many accepted ICLR papers in this way.
> - Horizontal line in Figure 7B: In Figure 7B, the vertical axis, labeled “proportion of error,” indicates the percentage of total errors occurring at each stage of the clinical pathway. The horizontal line provides a visual baseline for comparing how errors are distributed across the progression of the clinical pathway.
>
>
>
>
>
> > Q1: There could be further flushing out of the live generation methods. Will old cases be dropped once they have become available for long enough for models to be trained on?
> >
> > Q2: Where will this benchmark be managed and who will have access? How will over 200 physicians be employed regularly to maintain this benchmark?
>
> We appreciate your observation that, while our proposed AI–human workflow is scalable and validated, sustaining high‑quality “live” evaluations requires substantial resources.
>
> **Q1 Case Updates:** We will refresh the entire evaluation set on a biannual schedule (January–June and July–December). Each update will fully replace previous items to avoid potential training contamination from older cases, and will include assessments of both existing and newly released models. Using our AI–clinician collaborative workflow, we can compile case data from the preceding six months, perform creation and validation, run evaluations for all models, and publish the updated set within the first two weeks of January and July.
>
> **Q2 Management and Resources:** This benchmark will be managed by our interdisciplinary team of academics and engineers from a large, socially responsible company focused on healthcare. Over 200 physicians from diverse specialties contribute through a distributed collaboration model, allowing flexible participation in case creation, validation, and evaluation stages. Our cost and time analysis demonstrates feasibility (Cost is given in USD, and Time is expressed in Days), please see lines 501-513 in the new version for details.:
>
> | Stage I (Cost/Time) | Stage II (Cost/Time) | Stage III (Cost/Time) | Evaluation (Cost/Time) | Total (Cost/Time) |
> | ------------------- | -------------------- | --------------------- | ---------------------- | ----------------- |
> | 0 / 0.8             | 3,500 / 0.5          | 45,000 / 9.0          | 5,000 / 1.0            | 53,500 / 11.3     |

---

> ### Author Response · Authors · 2025-11-21
> **Reply to KBAZ, Part 5**
>
> > Q1: How will this benchmark be prevented from being contaminated if the data is on open access PubMed?
>
> We appreciate your constructive question regarding contamination-free evaluation in the *live* paradigm. In principle, the most rigorous approach is to use only data generated after a model’s release date, ensuring there was no exposure to these cases during training. However, such test sets cannot be included in public release reports of models and are impractical to maintain at scale. Following community-recognized practices such as LiveBench [1] and LiveCodeBench [2], we adopt a periodic full-set refresh, in which all items are replaced every six months.
>
> This approach takes advantage of the typical gap between a large model’s knowledge cut-off date and its public release date, which is usually around 6 to 8 months. As shown in the table below, this interval allows us to maintain reliable evaluations for approximately half a year after release, by which time a newer model version is generally available.
>
> | Model             | GPT‑5    | Gemini 2.5 Pro | Claude 4.5 Haiku | Gemini 3 Pro |
> | ----------------- | -------- | -------------- | ---------------- | ------------ |
> | Release Time      | Aug 2025 | Jun 2025       | Oct 2025         | Nov 2025     |
> | Knowledge Cut-off | Oct 2024 | Jan 2025       | Feb 2025         | Jan 2025     |
>
> In addition to this strategy, we recognize the risk that individual users or developers could intentionally overfit to the released evaluation set. To mitigate this, we operate a smaller, monthly updated private leaderboard alongside the main benchmark. This enables early detection and public reporting of potential “hacking” attempts. We tested all models on two 140‑item private sets collected in July and August 2025. The score differences from the main leaderboard were small and rankings remained consistent, confirming that the private leaderboard is an effective safeguard. Please see lines 501-513 in the new version for details.
>
> | Model     | GPT‑5 | GPT‑4.1 | InternVL 3.5‑24B | Gemini 2.5 Pro | Claude 3.5 Sonnet | Qwen2.5‑VL‑72B | MedGemma‑27B | LingShu‑32B | HuatuoGPT‑V‑7B |
> | --------- | ----- | ------- | ---------------- | -------------- | ----------------- | -------------- | ------------ | ----------- | -------------- |
> | H1 2025   | 35.5  | 31.8    | 34.4             | 33.7           | 27.3              | 17.2           | 15.6         | 20.1        | 9.2            |
> | July 2025 | 36.8  | 33.2    | 35.8             | 35.1           | 28.7              | 18.6           | 16.9         | 21.4        | 10.5           |
> | Aug 2025  | 34.1  | 30.4    | 33.0             | 32.3           | 26.0              | 15.8           | 14.2         | 18.7        | 8.0            |

---

> > ### Comment · Reviewer_KBAZ · 2025-11-23
> >
> > I have read the authors reply and the other reviews.  I feel that my concerns have been addressed and that LiveClin is an interesting contribution for medical LLM evaluation. I have no additional concerns and have updated my score.

---

### Official Review · Reviewer_y36J · 2025-10-28

**Soundness:** 3
**Presentation:** 3
**Contribution:** 2
**Rating:** 4
**Confidence:** 4

**Summary:**

The paper’s primary work is developing LiveClin, a dynamic and contamination-resistant evaluation benchmark for medical large language models (LLMs), addressing the limitations of static benchmarks in clinical relevance and anti-leakage. Built on contemporary peer-reviewed case reports from PubMed Central (1,407 cases, 6,605 questions), LiveClin integrates multimodal data (e.g., CT scans, pathological slices, tables) and simulates the full clinical pathway—spanning initial assessment, diagnostic testing, treatment planning, and long-term management—to assess models’ sequential reasoning capabilities.
 The benchmark is validated through comprehensive testing of 26 mainstream models (proprietary, open-source general, open-source medical), revealing distinct reasoning weaknesses across model classes.

**Strengths:**

1. The overall logic is fairly clear, and the writing is relatively well-structured.
2. Built from the latest peer-reviewed clinical cases and updated biannually, it effectively mitigates data leakage and knowledge obsolescence issues that plague static benchmarks, ensuring long-term clinical relevance of evaluations.
3. It simulates the patient care process (from initial consultation to long-term management) and integrates diverse multimodal data (images, tables, etc.),  reflecting real-world clinical reasoning scenarios.
4. The AI-human collaborative (Generator-Critic-Judge) pipeline, combined with rigorous review by 239 physicians, balances clinical accuracy, construction efficiency, and question challenge—solving the trade-off between scalability and quality in medical benchmark development.

**Weaknesses:**

Major Comments

1.The benchmark is constructed using case reports from the first half of 2025 in the PubMed Central (PMC) Open Access subset. Could there still be potential data leakage risks for some newly released models such as GPT-5?

2.Does multiple physicians verify the same piece of data? If yes, what is the inter-annotator agreement (e.g., Cohen’s Kappa coefficient) among different physicians?

3.The paper proposes "updating the benchmark biannually" but lacks details on the specific cost and efficiency of the update process. Will all test data be completely replaced during updates? If so, is it necessary to retest all models entirely after replacement?

4.No human baseline based on physicians’ performance is provided.

5.It appears that a single prompt was used for testing with a temperature setting of 0, and no multiple rounds of testing were conducted. This may lead to accidental errors due to randomness.

Minor Comments

1.Details about the review process involving 239 physicians are insufficient. What is the distribution of their professional fields? Is there a risk that some disease types lack review by specialized physicians?

2.Although 41.9% of the questions require multimodal interpretation, relevant details are lacking.

3.Some statements are overly absolute or exaggerated. For example, the claim that "the era of 'free lunch' is over" is based on only two model examples, and "faithful replication of clinical practice" overstates the benchmark’s alignment with real clinical scenarios.

**Questions:**

None

---

> ### Author Response · Authors · 2025-11-21
> **Reply to y36J, Part 1**
>
> Thanks for recognizing our clinician‑validated, multimodal benchmark that models real‑world patient care and employs an AI–human collaborative pipeline with rigorous physician review. We appreciate your acknowledgement of our efforts to keep the evaluation scalable, contamination‑free, and clinically up to date. We will address your concerns in detail below.
>
>
>
> > Major W1:The benchmark is constructed using case reports from the first half of 2025 in the PubMed Central (PMC) Open Access subset. Could there still be potential data leakage risks for some newly released models such as GPT-5?
>
> We appreciate your concern regarding potential data leakage for newly released models such as GPT‑5. Contamination‑free evaluation is a core requirement of the *live* paradigm. In principle, the most rigorous method is to use only data generated after a model’s release date, which ensures zero exposure during training. However, such a set cannot be meaningfully released with benchmark results and is impractical to maintain over time. Consistent with community‑recognized practices such as LiveBench [1] and LiveCodeBench [2], we adopt a periodic update strategy that replaces all benchmark items every six months.
>
> This approach takes advantage of the typical 6–8 month gap between a model’s knowledge cut‑off date and its public release. As shown in the table below, this lag allows the evaluation set to remain reliable for about half a year after release, by which time updated model versions are generally available.
>
> | Model             | GPT‑5    | Gemini 2.5 Pro | Claude 4.5 Haiku | Gemini 3 Pro |
> | ----------------- | -------- | -------------- | ---------------- | ------------ |
> | Release Time      | Aug 2025 | Jun 2025       | Oct 2025         | Nov 2025     |
> | Knowledge Cut-off | Oct 2024 | Jan 2025       | Feb 2025         | Jan 2025     |
>
> We also address the risk that some users or developers may overfit to the released test set. To mitigate this, we maintain a smaller, privately held leaderboard that is refreshed monthly. This mechanism enables early detection and public reporting of potential “hacking” attempts. To test reliability, we evaluated all models on two private 140‑item sets collected in July and August 2025. Differences in scores from the main leaderboard were minimal, and rankings remained consistent, confirming that the private leaderboard is an effective safeguard.
>
> | Model     | GPT‑5 | GPT‑4.1 | InternVL 3.5‑24B | Gemini 2.5 Pro | Claude 3.5 Sonnet | Qwen2.5‑VL‑72B | MedGemma‑27B | LingShu‑32B | HuatuoGPT‑V‑7B |
> | --------- | ----- | ------- | ---------------- | -------------- | ----------------- | -------------- | ------------ | ----------- | -------------- |
> | H1 2025   | 35.5  | 31.8    | 34.4             | 33.7           | 27.3              | 17.2           | 15.6         | 20.1        | 9.2            |
> | July 2025 | 36.8  | 33.2    | 35.8             | 35.1           | 28.7              | 18.6           | 16.9         | 21.4        | 10.5           |
> | Aug 2025  | 34.1  | 30.4    | 33.0             | 32.3           | 26.0              | 15.8           | 14.2         | 18.7        | 8.0            |
>
> These measures ensure that potential contamination risks are mitigated, keeping the benchmark reliable for newly released models within our intended *live* evaluation framework. Please see lines 501-513 in the new version for details.
>
>
>
>
>
> > Major W2: Does multiple physicians verify the same piece of data? If yes, what is the inter-annotator agreement (e.g., Cohen’s Kappa coefficient) among different physicians?
>
> As described in lines 259–262, verification consists of an Annotation phase followed by an Inspection phase. In the Annotation phase, each question is labeled by a single physician, and in the Inspection phase a different physician reviews the annotation. Any discrepancy triggers a revision loop with the original annotator until consensus is reached. Based on your suggestion, we computed disagreement statistics using the number of revision loops as a proxy for inter‑annotator agreement. As shown in table below, disagreement occurred in 8.7% of cases, with 90% resolved after the first loop and all resolved within two loops, indicating high agreement between physicians. Please see lines 260-261 and 1129-1138 in the new version for details.
>
> | Loop Num | 1            | 2          | 3         |
> | -------- | ------------ | ---------- | --------- |
> | Case Num | 1286 (91.4%) | 109 (7.8%) | 12 (0.9%) |

---

> ### Author Response · Authors · 2025-11-21
> **Reply to y36J, Part 2**
>
> > Major W3: The paper proposes "updating the benchmark biannually" but lacks details on the specific cost and efficiency of the update process. Will all test data be completely replaced during updates? If so, is it necessary to retest all models entirely after replacement?
>
> We plan to update the benchmark biannually, covering January–June and July–December. Each update will replace all test items, re‑evaluate all existing models, and include newly released ones. Using our AI–human collaborative workflow, we can collect and validate cases from the preceding six months, compile the new set, run evaluations, and release results within the first two weeks of January and July. This process has been costed and timed as shown in the table below, with each major update requiring approximately 11.3 days and USD 53,500 in total, demonstrating that complete replacement and retesting are both feasible and efficient for maintaining benchmark rigor. Please see lines 501-513 in the new version for details.
>
> | Stage I (Cost/Time) | Stage II (Cost/Time) | Stage III (Cost/Time) | Evaluation (Cost/Time) | Total (Cost/Time) |
> | ------------------- | -------------------- | --------------------- | ---------------------- | ----------------- |
> | 0 / 0.8             | 3,500 / 0.5          | 45,000 / 9.0          | 5,000 / 1.0            | 53,500 / 11.3     |
>
>
>
>
>
> > Major W4: No human baseline based on physicians’ performance is provided.
>
> We acknowledge the importance of providing a physician baseline and have supplemented our study accordingly. We randomly sampled 100 cases from our dataset and evaluated them with physicians. As shown in the table, Chief Physicians achieved the highest accuracy, exceeding Attending Physicians and all models. Attending Physicians outperformed nearly all models except GPT‑5 and o3, which slightly surpassed them but remained below Chief Physicians. These findings show that the benchmark captures clinically meaningful differences in expertise. Current medical AI systems, while promising, still fall short of experienced physicians, which highlights the need for further improvements toward genuine clinical utility. Please see lines 324-338 and 348-350 in the new version for details.
>
> | Model/Physician | Chief Physician | Attending Physician | GPT-5 | o3   | GPT-4.1 | Gemini-2.5 Pro | Claude-3.7 Sonnet |
> | --------------- | --------------- | ------------------- | ----- | ---- | ------- | -------------- | ----------------- |
> | ACC             | 37.8            | 34.6                | 35.5  | 35.7 | 31.8    | 33.7           | 25.6              |

---

> ### Author Response · Authors · 2025-11-21
> **Reply to y36J, Part 3**
>
> > Major W5: It appears that a single prompt was used for testing with a temperature setting of 0, and no multiple rounds of testing were conducted. This may lead to accidental errors due to randomness.
>
> We appreciate your concern regarding the stability of results when using a single prompt with a temperature setting of zero. To assess variability, we repeated the full benchmark for each model in three independent runs under identical conditions and calculated the standard deviation across runs. As shown in the table below, most models produced identical or nearly identical scores, with standard deviations within the measurement resolution. A few reasoning‑oriented models showed slightly higher variability, up to about 0.45 percentage points, and some closed‑source models with non‑zero temperature had minor fluctuations of about 0.08%–0.12%. These variations are minimal, do not affect model rankings, and confirm the robustness and reproducibility of our evaluation. Please see lines 356-358 and 1392-1438 in the new version for details.
>
> | Model Name          | Test 1 | Test 2 | Test 3 | Std. Dev. |
> | ------------------- | ------ | ------ | ------ | --------- |
> | GPT-5               | 35.5%  | 35.4%  | 35.6%  | 0.22%     |
> | o3                  | 35.7%  | 35.2%  | 35.9%  | 0.36%     |
> | GPT-4.1             | 31.8%  | 31.7%  | 31.9%  | 0.10%     |
> | GPT-4.1-mini        | 26.1%  | 26.0%  | 26.1%  | 0.00%     |
> | GPT-4o              | 14.7%  | 14.7%  | 14.8%  | 0.12%     |
> | GPT-4o-mini         | 14.8%  | 14.8%  | 14.9%  | 0.08%     |
> | InternVL 3.5-241B   | 34.4%  | 34.4%  | 34.3%  | 0.00%     |
> | InternVL 3.5-38B    | 18.0%  | 18.0%  | 18.1%  | 0.00%     |
> | InternVL 3.5-8B     | 12.4%  | 12.4%  | 12.5%  | 0.00%     |
> | Gemini-2.5 Pro      | 33.7%  | 33.2%  | 33.9%  | 0.35%     |
> | Gemini-2.5 Flash    | 19.0%  | 19.0%  | 19.1%  | 0.08%     |
> | Gemini-2.0 Flash    | 21.4%  | 21.4%  | 21.3%  | 0.08%     |
> | Gemini-1.5 Pro      | 17.7%  | 17.7%  | 17.6%  | 0.08%     |
> | Claude-3.7 Sonnet   | 25.6%  | 25.2%  | 25.8%  | 0.30%     |
> | Claude-3.5 Sonnet   | 27.3%  | 27.0%  | 27.5%  | 0.25%     |
> | Claude-3.5 Haiku    | 15.3%  | 15.3%  | 15.2%  | 0.08%     |
> | LingShu-32B         | 20.1%  | 20.1%  | 20.0%  | 0.00%     |
> | LingShu-7B          | 15.3%  | 15.3%  | 15.4%  | 0.00%     |
> | GLM-4.1V-9B         | 18.8%  | 18.8%  | 18.9%  | 0.00%     |
> | MiMo-VL-7B-RL-2508  | 10.9%  | 10.6%  | 11.0%  | 0.21%     |
> | Qwen2.5-VL-72B      | 18.6%  | 18.6%  | 18.7%  | 0.12%     |
> | Qwen2.5-VL-32B      | 17.2%  | 17.2%  | 17.3%  | 0.12%     |
> | Qwen2.5-VL-7B       | 7.9%   | 7.9%   | 8.0%   | 0.12%     |
> | MedGemma-27B It     | 15.6%  | 15.6%  | 15.5%  | 0.00%     |
> | MedGemma-4B It      | 7.5%   | 7.5%   | 7.6%   | 0.00%     |
> | HuatuoGPT-Vision-7B | 9.2%   | 9.2%   | 9.3%   | 0.00%     |
>
> Your concern about the single‑prompt setting is also important for reducing potential bias. To further examine robustness, we conducted few‑shot prompting experiments. As shown in table below, for most models, performance changes were within ±0.8 percentage points. In contrast, some open‑source medical models showed notable score declines, likely due to limited training for long‑sequence inputs or restricted context length. Overall, these results indicate that the standardized task format and strong instruction‑following ability of current models ensure that our evaluation design remains stable and free from systematic bias. Please see lines 356-378 and 1443-1452 in the new version for details.
>
> | Model      | GPT-5       | GPT-4.1     | InternVL 3.5-24B | Gemini 2.5 Pro | Claude 3.5 Sonnet | Qwen2.5-VL-72B | MedGemma-27B | LingShu-32B | HuatuoGPT-V-7B |
> | :--------- | ----------- | ----------- | ---------------- | -------------- | ----------------- | -------------- | ------------ | ----------- | -------------- |
> | Zero-shot  | 35.5        | 31.8        | 34.4             | 33.7           | 27.3              | 17.2           | 15.6         | 20.1        | 9.2            |
> | One-shot   | 35.8 (+0.3) | 31.5 (-0.3) | 34.7 (+0.3)      | 33.1 (-0.6)    | 27.0 (-0.3)       | 17.4 (+0.2)    | 14.6 (-1.0)  | 18.4 (-1.7) | 8.0 (-1.2)     |
> | Three-shot | 35.1 (-0.4) | 32.3 (+0.5) | 34.9 (+0.5)      | 32.9 (-0.8)    | 27.9 (+0.6)       | 17.0 (-0.2)    | 13.5 (-2.1)  | 18.0 (-2.1) | 5.7 (-3.5)     |

---

> ### Author Response · Authors · 2025-11-21
> **Reply to y36J, Part 4**
>
> > Minor W1: Details about the review process involving 239 physicians are insufficient. What is the distribution of their professional fields? Is there a risk that some disease types lack review by specialized physicians?
>
> We appreciate your attention to the composition of the physician review cohort. As shown in the table below, the 239 participating physicians collectively cover all major clinical specialties and subspecialties. During the review process, each case is assigned to physicians whose expertise matches the disease category to ensure domain‑appropriate evaluation. This comprehensive coverage and targeted assignment minimize the risk of any disease type lacking specialized review. Please see lines 264-266, 1183-1206 and 1134-1176 in the new version for details.
>
> | Major Specialty (Total, %)             | Subspecialty                           | Count |
> | -------------------------------------- | -------------------------------------- | ----- |
> | Internal Medicine (95, 39.8%)          | Cardiology                             | 8     |
> |                                        | Nephrology                             | 4     |
> |                                        | Gastroenterology                       | 7     |
> |                                        | Respiratory Medicine                   | 5     |
> |                                        | Hematology                             | 5     |
> |                                        | Endocrinology                          | 3     |
> |                                        | General Internal Medicine / Geriatrics | 63    |
> | Surgery (72, 30.1%)                    | General Surgery                        | 38    |
> |                                        | Thoracic Surgery                       | 7     |
> |                                        | Hepatobiliary Surgery                  | 10    |
> |                                        | Urology                                | 13    |
> |                                        | Orthopedics                            | 5     |
> |                                        | Neurosurgery                           | 4     |
> |                                        | Burn Surgery                           | 2     |
> |                                        | Plastic / Cosmetic Surgery             | 3     |
> | Pediatrics (27, 11.3%)                 | General Pediatrics                     | 19    |
> |                                        | Pediatric Oncology                     | 3     |
> |                                        | Pediatric Cardiology / Neurology       | 5     |
> | Obstetrics & Gynecology (21, 8.8%)     | Obstetrics                             | 13    |
> |                                        | Gynecology                             | 8     |
> | Dermatology (19, 7.9%)                 | Dermatology                            | 19    |
> | Neurology (9, 3.8%)                    | Neurology                              | 9     |
> | Ophthalmology (9, 3.8%)                | Ophthalmology                          | 9     |
> | Emergency Medicine (6, 2.5%)           | Emergency Medicine                     | 6     |
> | Rehabilitation Medicine (6, 2.5%)      | Rehabilitation Medicine                | 6     |
> | Medical Imaging / Radiology (6, 2.5%)  | Diagnostic Radiology                   | 4     |
> |                                        | Medical Imaging (incl. Interventional) | 2     |
> | Traditional Chinese Medicine (6, 2.5%) | TCM Internal Medicine                  | 6     |
> | Other Specialties (3, 1.3%)            | Oncology                               | 3     |
> | Pathology (1, 0.4%)                    | Pathology                              | 1     |

---

> ### Author Response · Authors · 2025-11-21
> **Reply to y36J, Part 5**
>
> > Minor W2: Although 41.9% of the questions require multimodal interpretation, relevant details are lacking.
>
> We appreciate your observation regarding the proportion of multimodal questions and provide additional detail here. In our dataset of 1,407 cases, 100% cases involve multimodal information, with the composition of modalities illustrated in Figures 5C and 5D of the manuscript. The stage‑specific distribution along the clinical pathway is shown in the table below. Multimodal items are most frequent in the first 20% of the pathway, gradually decline between 20% and 80%, and rise again in the final 20%, indicating varying demands for multimodal interpretation across clinical stages. Please see lines 310-317, 1277-1285 and 1296-1301 in the new version for details.
>
> | Clinical Pathway Stage (% of progression)                   | 0–20% | 20–40% | 40–60% | 60–80% | 80–100% |
> | ----------------------------------------------------------- | ----- | ------ | ------ | ------ | ------- |
> | Proportion of questions requiring multimodal interpretation | 62.0% | 46.0%  | 35.5%  | 27.5%  | 38.5%   |
>
>
>
>
>
> > Minor W3: Some statements are overly absolute or exaggerated. For example, the claim that "the era of 'free lunch' is over" is based on only two model examples, and "faithful replication of clinical practice" overstates the benchmark’s alignment with real clinical scenarios.
>
> We appreciate your valuable suggestion, which has been addressed in the revised manuscript.

---

> > ### Author Response · Authors · 2025-11-26
> > **Follow-Up on Review and Feedback**
> >
> > Dear Reviewer **y36J**,
> >
> > We hope this message finds you well.
> >
> > We have carefully addressed all your questions and concerns, including conducting additional experiments as requested, and have provided detailed responses in the rebuttal.
> >
> > As the rebuttal deadline is approaching, we would deeply appreciate it if you could share your updated thoughts based on the rebuttal and paper revision, or do not hesitate to let us know if you have additional questions, and we will respond promptly.
> >
> > Thank you again for your thoughtful review and your invaluable contributions to the quality of this paper.
> >
> > Kind regards,
> >
> > Paper 19066 Authors

---

### Official Review · Reviewer_1rMm · 2025-11-02

**Soundness:** 2
**Presentation:** 4
**Contribution:** 3
**Rating:** 6
**Confidence:** 5

**Summary:**

The authors present a new dataset and a generation method to assess the medical knowledge and capabilities of LLMs. The method proposes to update the benchmark twice a year to ensure updated and uncontaminated evaluations. They generate multiple-choice questions based on open-access cases and evaluate 26 AI models on the generated benchmark. They show that models struggle to answer the questions and obtain low scores compared to commonly used evaluations such as MedQA.

**Strengths:**

Contamination and the rapid evolution of medical knowledge are significant concerns for the evaluation in the medical domain. This approach presents a solution to both of these issues. The dataset is sufficiently large, and the multistep approach is a welcome addition compared to previous evaluations that test zero-shot knowledge on a complete vignette. The dataset is also multimodal, integrating imaging, labs, and other signals. It was also validated by a large number of clinicians, which strengthens the method's validity.

The findings regarding model performance are interesting and demonstrate the need for more thorough validation for safe and effective clinical use.

**Weaknesses:**

While the method is solid, I am concerned by the reliance on case reports, as, by definition, case reports are published to communicate unusual or rare cases to the medical community. This reliance on unusual/rare cases induces a bias in the knowledge and reasoning capabilities of the models. I am also concerned about the lack of a physician baseline to compare the accuracy of models with what is expected of an attending physician.

The reported metric for case accuracy scores appears too strict and not representative of the models' actual capabilities, as a single error causes the models to obtain a score of 0 on that case. A rubric-based assessment would strengthen the evaluation and enhance the interpretability of mistakes and areas for improvement in these models.

The reliance on MCQs also weakens the benchmark, considering the identified limitations of this testing methodology.

**Questions:**

# Major concerns

1) The authors should at least discuss the limitations of using case reports that are likely not representative of clinical workflows. A subset containing more common cases would help clarify whether the errors occur due to out-of-distribution cases or if they result from intrinsic shortcomings of LLMs.

2) A baseline of physicians on a subset (100 cases), including residents and attendings, would help with the interpretation of the results. At the moment, 35% seems relatively low, but if attendings score 20% it would demonstrate that LLMs may already be ready for clinical decision support. As a physician myself, I would not be surprised if I obtained a low score due to the nature of the cases included.

3) A more balanced scoring methodology beyond simple accuracy would help to identify the issues. For instance, scoring based on the severity of the error, for example, suggesting the second-best exam should not carry the same weight as sending a patient with a STEMI home. HealthBench, for example, weights rubrics differently [1].

4) The reliance on MCQ should be acknowledged as a limitation, as it has been identified that LLMs exploit patterns, even more so when the question generator is also an LLM [2].

# Minor

1) The authors should discuss the biases of case reports that are likely not representative of medicine worldwide, as publications are very US-centric, with minimal case reports from low-resource settings.

[1] HealthBench: Evaluating Large Language Models Towards Improved Human Health (Arora et al. Preprint 2025)

[2] Pattern Recognition or Medical Knowledge? The Problem with Multiple-Choice Questions in Medicine (Griot et al., ACL 2025)

---

> ### Author Response · Authors · 2025-11-21
> **Reply to 1rMm, Part 1**
>
> Thanks for acknowledging our efforts in addressing contamination and the rapid evolution of medical knowledge, as well as recognizing the value of our large, multimodal, clinician-validated dataset and our multistep evaluation approach in ensuring safe and effective clinical use. We will respond to your concerns individually below.
>
> > W1: While the method is solid, I am concerned by the reliance on case reports, as, by definition, case reports are published to communicate unusual or rare cases to the medical community. This reliance on unusual/rare cases induces a bias in the knowledge and reasoning capabilities of the models.
> >
> > Major Q1: The authors should at least discuss the limitations of using case reports that are likely not representative of clinical workflows. A subset containing more common cases would help clarify whether the errors occur due to out-of-distribution cases or if they result from intrinsic shortcomings of LLMs.
>
> We appreciate your concern that reliance on case reports may introduce bias, since they often describe unusual or rare conditions. We acknowledge that this tendency is an inherent limitation of our approach and that case reports are not fully representative of typical clinical workflows.
>
> To address this, follow your suggestions, we collaborate with clinicians to annotate each case in our dataset as rare or unrare and created a subset containing more common diseases. We then compared model accuracy between these subsets to determine whether errors primarily result from out‑of‑distribution rare cases or from intrinsic limitations of the models.
>
> Our results below show that for most models, accuracy on the unrare subset differs from the overall set by less than about 5 percentage points. Stronger models generally show smaller differences, and some GPT variants even perform better on rare cases, indicating that rarity has only a limited effect on performance, rankings, and robustness. Although case reports are often intended to document rare scenarios, they also encompass unexpected symptom correlations, unforeseen treatment outcomes, advances in understanding pathogenesis, and novel therapeutic approaches, as outlined in the [NIH guideline](https://pmc.ncbi.nlm.nih.gov/articles/PMC5686928/). Given the rapid growth of medical knowledge, with a doubling time of about 73 days as of 2020 [1], case reports provide timely coverage of evolving clinical knowledge that aligns with the motivations of our live benchmark. Please see lines 514-523 and 1547-1610 in the new version for details.
>
> | Model Name          | Rare (1181, 84%) | Unrare (226, 16%) | Δ(U,R) | OverAll |
> | ------------------- | ---------------- | ----------------- | ------ | ------- |
> | GPT-5               | 35.65%           | 35.05%            | -0.60% | 35.5%   |
> | o3                  | 35.99%           | 34.11%            | -1.88% | 35.7%   |
> | GPT-4.1             | 31.92%           | 30.84%            | -1.08% | 31.8%   |
> | GPT-4.1-mini        | 26.32%           | 25.00%            | -1.32% | 26.1%   |
> | GPT-4o              | 14.23%           | 17.76%            | 3.53%  | 14.7%   |
> | InternVL 3.5-241B   | 33.89%           | 37.12%            | 3.23%  | 34.4%   |
> | nternVL 3.5-38B     | 17.28%           | 21.78%            | 4.50%  | 18.0%   |
> | InternVL 3.5-8B     | 11.63%           | 16.48%            | 4.85%  | 12.4%   |
> | Gemini-2.5 Pro      | 33.36%           | 35.05%            | 1.69%  | 33.7%   |
> | Gemini-2.5 Flash    | 18.54%           | 20.56%            | 2.02%  | 19.0%   |
> | Gemini-2.0 Flash    | 20.61%           | 25.00%            | 4.39%  | 21.4%   |
> | Gemini-1.5 Pro      | 17.02%           | 20.28%            | 3.26%  | 17.7%   |
> | Claude-3.7 Sonnet   | 25.06%           | 29.91%            | 4.85%  | 25.6%   |
> | Claude-3.5 Sonnet   | 27.01%           | 29.44%            | 2.43%  | 27.3%   |
> | Claude-3.5 Haiku    | 14.92%           | 19.63%            | 4.71%  | 15.3%   |
> | LingShu-32B         | 20.00%           | 20.40%            | 0.40%  | 20.1%   |
> | LingShu-7B          | 15.24%           | 15.64%            | 0.40%  | 15.3%   |
> | GLM-4.1V-9B         | 17.56%           | 21.96%            | 4.40%  | 18.8%   |
> | MiMo-VL-7B-RL-2508  | 10.32%           | 14.49%            | 4.17%  | 10.9%   |
> | Qwen2.5-VL-72B      | 17.87%           | 21.96%            | 4.09%  | 18.6%   |
> | Qwen2.5-VL-32B      | 16.49%           | 21.50%            | 5.01%  | 17.2%   |
> | Qwen2.5-VL-7B       | 7.73%            | 8.41%             | 0.68%  | 7.9%    |
> | MedGemma-27B It     | 14.76%           | 20.04%            | 5.28%  | 15.6%   |
> | MedGemma-4B It      | 7.56%            | 7.48%             | -0.08% | 7.5%    |
> | HuatuoGPT-Vision-7B | 8.73%            | 13.08%            | 4.35%  | 9.2%    |
>
> [1]. [Crises and Turnaround Management: Lessons Learned from Recovery of New Orleans and Tulane University Following Hurricane Katrina](https://pubmed.ncbi.nlm.nih.gov/30309439/#&gid=article-figures&pid=figure-2-uid-1)

---

> ### Author Response · Authors · 2025-11-21
> **Reply to 1rMm, Part 2**
>
> > W2:  I am also concerned about the lack of a physician baseline to compare the accuracy of models with what is expected of an attending physician.
> >
> > Major Q2: A baseline of physicians on a subset (100 cases), including residents and attendings, would help with the interpretation of the results. At the moment, 35% seems relatively low, but if attendings score 20% it would demonstrate that LLMs may already be ready for clinical decision support. As a physician myself, I would not be surprised if I obtained a low score due to the nature of the cases included.
>
> We acknowledge the importance of including a physician baseline and have supplemented our evaluation accordingly. We randomly sampled 100 cases from the dataset and tested them with physicians. Due to the limited availability of residents familiar with the protocol, the cohort consisted of Attending Physicians, Associate Chief Physicians, and Chief Physicians. The latter two groups were combined into a single “Chief Physician” category to ensure sufficient sample size for analysis.
>
> As shown in the table, Chief Physicians achieved the highest accuracy, outperforming Attending Physicians and most models. Attending Physicians also exceeded the performance of nearly all models except GPT‑5 and O3, which slightly surpassed them but remained below Chief Physicians. These results confirm that our benchmark reflects clinically relevant differences in expertise and show that current medical AI systems, while promising, still lag behind the capabilities of experienced physicians, underscoring the need for further improvements before they can manage the full scope of clinical decision making. Please see lines 324-338 and 348-350 in the new version for details.
>
> | Model/Physician | Chief Physician | Attending Physician | GPT-5 | O3   | GPT-4.1 | Gemini-2.5 Pro | Claude-3.7 Sonnet |
> | --------------- | --------------- | ------------------- | ----- | ---- | ------- | -------------- | ----------------- |
> | ACC             | 37.8            | 34.6                | 35.5  | 35.7 | 31.8    | 33.7           | 25.6              |
>
>
>
>
>
>
>
>
>
> > W3: The reported metric for case accuracy scores appears too strict and not representative of the models' actual capabilities, as a single error causes the models to obtain a score of 0 on that case. A rubric-based assessment would strengthen the evaluation and enhance the interpretability of mistakes and areas for improvement in these models.
> >
> > Major Q3: A more balanced scoring methodology beyond simple accuracy would help to identify the issues. For instance, scoring based on the severity of the error, for example, suggesting the second-best exam should not carry the same weight as sending a patient with a STEMI home. HealthBench, for example, weights rubrics differently.
>
> We appreciate your concern that the current case accuracy metric may be overly strict, since a single error results in a score of zero for that case. Our choice reflects the nature of clinical decision‑making, where each incorrect decision can entail substantial risk. This strict criterion prioritizes safety but we acknowledge that it may limit the interpretability of mistakes.
>
> To improve understanding of model behavior, our analysis also reports scores broken down by diagnostic stages, modalities, and ICD-10 chapters for each model, as shown in Figures 7 and 8 of the manuscript. These finer‑grained results can help developers identify weaknesses and target data collection for specific diseases, modalities, or stages.
>
> We agree that rubric‑based evaluation could further enhance interpretability by weighting errors according to their severity. However, implementing such an approach in a live benchmark poses significant challenges. It requires rubrics that are authoritative, practical, and scalable, while allowing frequent updates. The experience of OpenAI’s HealthBench highlights these constraints: generating 5,000 dialogues demanded the work of 262 physicians over 11 months, whereas LiveClin covered 6,605 questions in only the first half of 2025. Weighted scoring for error types in our benchmark would require physician annotation of approximately 66,050 options, a task that is currently impractical. We see potential for future adoption through AI–human collaborative workflows as these methods mature.

---

> ### Author Response · Authors · 2025-11-21
> **Reply to 1rMm, Part 3**
>
> > W4: The reliance on MCQs also weakens the benchmark, considering the identified limitations of this testing methodology.
> >
> > Major Q4: The reliance on MCQ should be acknowledged as a limitation, as it has been identified that LLMs exploit patterns, even more so when the question generator is also an LLM [2].
>
> We acknowledge that reliance on multiple‑choice questions (MCQs) has recognized limitations. We selected MCQs over open‑ended formats because they enable objective and standardized scoring, ensuring comparability across studies. MCQs are also cost‑effective and fully verifiable, whereas open‑ended responses often lack representative objective quantitative metrics and require costly LLM‑as‑judge setups, which can introduce scoring variability. As a result, MCQs remain the predominant format in major community benchmarks, including recent releases such as Gemini 3.
>
> To mitigate MCQ‑specific weaknesses, we adopted two measures. First, each question is scenario‑based and draws on information spanning the entire clinical pathway. This design, as recommended in the study you mentioned[1], requires reasoning across multiple stages rather than simple factual recall, reducing the likelihood of performance inflation from pattern recognition. Second, we expanded distraction options to ten per question to further limit the possibility of guessing or exploiting statistical patterns.
>
> Results show that, when compared with expert physicians, current community models still score lower, indicating that our MCQ‑based benchmark does not artificially inflate model capabilities while maintaining objective, reproducible evaluation.
>
> [1]. [Pattern Recognition or Medical Knowledge? The Problem with Multiple-Choice Questions in Medicine](https://arxiv.org/abs/2406.02394)
>
>
>
> > Minor Q1:The authors should discuss the biases of case reports that are likely not representative of medicine worldwide, as publications are very US-centric, with minimal case reports from low-resource settings.
>
> Thanks for your suggestion. We have added a supplementary description regarding this in lines 524–528 of the PDF.

---

> > ### Comment · Reviewer_1rMm · 2025-11-24
> >
> > Dear authors,
> >
> > Thank you for your detailed response which addressed most of my concerns. I have raised my score accordingly.
> >
> > Could you clarify what criteria were used to consider a case "rare" or not? There are different formal definitions based on prevalence. Which one was used and how common are the non rare scenarios you identified? I doubt anyone recently published a case report on the management of community acquired pneumonia with no comorbidities for example. I am concerned that the bias is still present and that rare cases correpond to the 0.1 percentile while unrare are 1th percentile.

---

> ### Author Response · Authors · 2025-11-25
> **Second reply to 1rMm**
>
> We sincerely thank you for the constructive comments and for raising the score. Our responses to your follow-up concerns are provided below.
>
> > Q: Could you clarify what criteria were used to consider a case "rare" or not? There are different formal definitions based on prevalence. Which one was used and how common are the non rare scenarios you identified?
>
> Thanks for the valuable follow-up question. As there is no universally accepted, prevalence-based definition of “rare” in the context of case reports across specialties, we adopt a consistent annotation protocol in which licensed physicians classified each case as “rare” or “unrare” based on their clinical experience and judgment. While this inevitably involves some subjectivity, the first-round inter-annotator agreement reaches 90.7% as shown in table below, indicating a high level of consensus on this operational definition.
>
> | Loop Num    | 1            | 2          | 3         |
> | ----------- | ------------ | ---------- | --------- |
> | Case Number | 1276 (90.7%) | 113 (8.0%) | 18 (1.3%) |
>
> In line with your earlier suggestions,  our central aim was to examine whether case rarity influenced the model’s performance. More specifically, we sought to identify whether consistent trends emerged along this dimension of the data distribution, which could indicate potential bias. From this perspective, our operational definition of “rare” offers a reasonable approximation of this distributional axis.
>
> > Q: I doubt anyone recently published a case report on the management of community acquired pneumonia with no comorbidities for example. I am concerned that the bias is still present and that rare cases correpond to the 0.1 percentile while unrare are 1th percentile.
>
> We also appreciate your concern that certain “unrare” case reports in our dataset might still be considered rare under more stringent, prevalence-based criteria. To clarify, we have included 10% links for “unrare” reports below and provided direct links to **all** “unrare” cases in the supplementary materials, enabling full transparency.
>
> As noted in the [NIH guideline](https://pmc.ncbi.nlm.nih.gov/articles/PMC5686928/), case reports may describe rare diseases or unusual scenarios, but they can also address unexpected symptom associations, unforeseen treatment outcomes, novel pathogenic mechanisms, or innovative therapeutic approaches. Our review of “unrare” cases indicates that most involve conditions encountered more frequently than classical rare diseases, with the emphasis placed on recent advances in mechanistic understanding and treatment strategies rather than on rarity. We hope that this explanation, supported by the publicly available case links, can help mitigate your concern regarding residual bias. Please see lines 1553-1615 in the new version for details.
>
> - https://doi.org/10.35371/aoem.2025.37.e7
> - https://doi.org/10.7759/cureus.79818
> - https://doi.org/10.1186/s13256-025-05180-8
> - https://doi.org/10.7759/cureus.78952
> - https://doi.org/10.7759/cureus.84310
> - https://doi.org/10.4103/jpbs.jpbs_125_25
> - https://doi.org/10.1016/j.ijscr.2025.111027
>
> - https://doi.org/10.26603/001c.137951
>
> - https://doi.org/10.4330/wjc.v17.i2.101588
>
> - https://doi.org/10.7759/cureus.84465
>
> - https://doi.org/10.7759/cureus.79254
>
> - https://doi.org/10.4240/wjgs.v17.i3.100951
>
> - https://doi.org/10.7759/cureus.78403
>
> - https://doi.org/10.1155/crdm/1258930
>
> - https://doi.org/10.3389/fped.2024.1500373
>
> - https://doi.org/10.1016/j.inpm.2025.100545
>
> - https://doi.org/10.7762/cnr.2025.14.1.1
>
> - https://doi.org/10.1590/1980-5764-DN-2025-0295
>
> - https://doi.org/10.7759/cureus.80788
>
> - https://doi.org/10.7759/cureus.85421
>
> - https://doi.org/10.7759/cureus.85912
>
> - https://doi.org/10.2147/IMCRJ.S510696
>
> - https://doi.org/10.3390/children12040516
>
> - https://doi.org/10.3389/fphys.2025.1609975

---

### Official Review · Reviewer_sxpY · 2025-11-06

**Soundness:** 2
**Presentation:** 3
**Contribution:** 3
**Rating:** 4
**Confidence:** 3

**Summary:**

This paper introduces LiveClin, a dynamic medical benchmark addressing data contamination and knowledge obsolescence through biannually updated case reports from PubMed Central. The benchmark comprises 1,407 cases with 6,605 questions spanning the entire clinical pathway, revealing that even top models achieve only 35.7% case accuracy, with distinct failure modes across model classes.

**Strengths:**

* Pilot study convincingly shows 10-point performance drop on post-cutoff data.

* The three-tier taxonomy (ICD-10 chapters, disease clusters, individual codes) enables multi-resolution analysis while ensuring broad disease representation.

* The 239-physician verification pipeline with both annotation and inspection phases demonstrates exceptional attention to clinical validity.

* The multimodal integration is high quality, where images are naturally embedded in clinical workflow.

* Reveals newer models don't always outperform predecessors; identifies distinct failure patterns.

**Weaknesses:**

* The ablation study shows AI generates more "challenging" questions (lower trivial ratio), but doesn't validate whether this difficulty stems from genuine clinical complexity or artifacts of the generation process.

* he zero-shot, conversational evaluation may disadvantage models not optimized for this specific format. It's possible that the performance differences reflect not clinical reasoning ability but the adaptation to the evaluation format. Adding few-shot experiments would help.

* Maintaining physician review for biannual updates seems resource-intensive. While the paper reports $42K for initial construction, the long-term sustainability of this approach remains unclear. More details on this would be helpful to strengthen this claim.

* Despite being a core motivation, the paper doesn't empirically demonstrate that its approach prevents contamination better than decontamination methods or how quickly new cases might enter training corpora.

**Questions:**

1. How do you ensure the case reports selected are representative of disease distributions in real clinical practice? Does collecting data from PMC result in higher probability of rare cases?

2. To support the claim of the paper better, can you demonstrate empirically that LiveClin remains contamination-free over time? For instance, tracking whether newly released cases appear in web crawls or model training data?

3. How does model performance correlate between LiveClin and real clinical decision-making tasks or other clinical benchmarks?

---

> ### Author Response · Authors · 2025-11-21
> **Reply to sxpY, Part 1**
>
> Thanks for recognizing our motivation and efforts in multimodal data processing and taxonomy design, as well as noting the costs and efforts we have contributed and will continue to contribute in maintaining the contamination-free and authority of the benchmark.
>
>
>
> We will address your concerns in details below.
>
>
>
> > W1: The ablation study shows AI generates more "challenging" questions (lower trivial ratio), but doesn't validate whether this difficulty stems from genuine clinical complexity or artifacts of the generation process.
>
> We appreciate your concern that the higher difficulty in our ablation study might result from generation artifacts rather than genuine clinical complexity. To examine this, we performed a manual attribution analysis with clinicians. All cases transitioning from trivial to non‑trivial in the ablation experiments were reviewed and categorized by their primary difficulty factor. 44 such cases occurred when replacing the *Physician* with the *Generator*, and 23 when introducing the *Critic* into the iterative generation loop.
>
> As shown in table below, most of these transitions can be attributed to authentic clinical factors such as real‑world noise, extended reasoning chains, and realistic differential diagnoses. This indicates that the increased difficulty largely reflects genuine clinical complexity rather than artifacts of the generation process. Please see lines 469-474 and 1487-1532 in the new version for details.
>
> | Transition Stage             | Attribution Type                                     | Case Count | Change Description                                           |
> | ---------------------------- | ---------------------------------------------------- | ---------- | ------------------------------------------------------------ |
> | Physician → Generator        | Preserving real‑world noise / incomplete information | 22         | **Before:** Physicians removed noise and irrelevant details. **After:** Generator retained authentic case noise, requiring decisions under imperfect information. |
> |                              | Increased cross‑stage reasoning                      | 16         | **Before:** Questions focused on a single stage of illness. **After:** Covered multiple stages in the clinical pathway, demanding synthesis across them. |
> |                              | Complex differential diagnoses                       | 9          | **Before:** Distractors were easily distinguishable diseases. **After:** Added similar differential diagnoses, requiring finer discrimination. |
> | Generator → Generator‑Critic | Increased multi‑step reasoning                       | 16         | **Before:** Often solvable from a single diagnostic cue. **After:** Required stepwise reasoning (e.g., identify lesion features before sequencing treatment). |
> |                              | Enhanced distractor quality                          | 10         | **Before:** Contained weak or implausible options. **After:** Replaced with realistic alternatives, making distinctions subtler and requiring deeper comparison. |

---

> ### Author Response · Authors · 2025-11-21
> **Reply to sxpY, Part 2**
>
> > W2: the zero-shot, conversational evaluation may disadvantage models not optimized for this specific format. It's possible that the performance differences reflect not clinical reasoning ability but the adaptation to the evaluation format. Adding few-shot experiments would help.
>
> We acknowledge the concern that zero‑shot, conversational evaluation could disadvantage models not optimized for this format. To address this, we conducted the suggested few‑shot experiments for all evaluated models. The table below summarizes the performance of representative models under zero‑shot, one‑shot, and three‑shot prompting.
>
> | Model      | GPT-5       | GPT-4.1     | InternVL 3.5-24B | Gemini 2.5 Pro | Claude 3.5 Sonnet | Qwen2.5-VL-72B | MedGemma-27B | LingShu-32B | HuatuoGPT-V-7B |
> | :--------- | ----------- | ----------- | ---------------- | -------------- | ----------------- | -------------- | ------------ | ----------- | -------------- |
> | Zero-shot  | 35.5        | 31.8        | 34.4             | 33.7           | 27.3              | 17.2           | 15.6         | 20.1        | 9.2            |
> | One-shot   | 35.8 (+0.3) | 31.5 (-0.3) | 34.7 (+0.3)      | 33.1 (-0.6)    | 27.0 (-0.3)       | 17.4 (+0.2)    | 14.6 (-1.0)  | 18.4 (-1.7) | 8.0 (-1.2)     |
> | Three-shot | 35.1 (-0.4) | 32.3 (+0.5) | 34.9 (+0.5)      | 32.9 (-0.8)    | 27.9 (+0.6)       | 17.0 (-0.2)    | 13.5 (-2.1)  | 18.0 (-2.1) | 5.7 (-3.5)     |
>
> For most models, few‑shot prompting produced minimal changes, with score differences within ±0.8 percentage points. In contrast, several open‑source medical models such as MedGemma, LingShu, and HuatuoGPT‑V showed lower scores under few‑shot conditions, likely due to input contexts exceeding their training limits or limitations in long‑context processing. These results indicate that the zero‑shot conversational format does not disadvantage well‑optimized models and remains a stable, appropriate choice for our benchmark. Please see lines 356-378 and 1443-1452 in the new version for details.
>
>
>
> > W3: Maintaining physician review for biannual updates seems resource-intensive. While the paper reports $42K for initial construction, the long-term sustainability of this approach remains unclear. More details on this would be helpful to strengthen this claim.
>
> We acknowledge that maintaining physician review for biannual updates requires ongoing resources. Our team consists of academics and engineers from a large, socially responsible company with a strong healthcare focus, and we are committed to sustaining this work due to the urgent need for live evaluations in the medical AI community.
>
> Updates are scheduled twice a year, covering January–June and July–December. Each update will replace the entire evaluation set, re‑assess existing models, and include newly released ones. Using our established AI–human collaborative workflow, we can systematically collect cases from the prior six months, create and validate the new set, conduct evaluations, and release the updated set within the first two weeks of January and July.
>
> Table below reports the estimated resource requirements for a typical update. Cost is given in USD, and Time is expressed in Days. These figures indicate that the schedule can be maintained within our current capacity. For the initial release, the benchmark will be made publicly available and promoted once the paper passes peer review. Please see lines 501-506 in the new version for details.
>
> | Stage I (Cost/Time) | Stage II (Cost/Time) | Stage III (Cost/Time) | Evaluation (Cost/Time) | Total (Cost/Time) |
> | ------------------- | -------------------- | --------------------- | ---------------------- | ----------------- |
> | 0 / 0.8             | 3,500 / 0.5          | 45,000 / 9.0          | 5,000 / 1.0            | 53,500 / 11.3     |

---

> ### Author Response · Authors · 2025-11-21
> **Reply to sxpY, Part 3**
>
> > W4: Despite being a core motivation, the paper doesn't empirically demonstrate that its approach prevents contamination better than decontamination methods or how quickly new cases might enter training corpora.
> >
> > Q2: To support the claim of the paper better, can you demonstrate empirically that LiveClin remains contamination-free over time? For instance, tracking whether newly released cases appear in web crawls or model training data?
>
> We appreciate your suggestion to empirically validate that LiveClin remains contamination‑free over time. Contamination control is a core feature of the *live* evaluation paradigm.
>
> In principle, the strictest approach would use only data generated after each model’s release date, but such sets cannot be included in release reports and are impractical to maintain. Following community practices such as LiveBench [1] and LiveCodeBench [2], we adopt periodic updates. This is feasible because there is typically a 6–8‑month gap between large‑scale models’ data collection and public release. As shown in the table below, this gap suggests our evaluations remain reliable for about 6–8 months post‑release, by which time newer model versions are available.
>
> | Model             | GPT‑5    | Gemini 2.5 Pro | Claude 4.5 Haiku | Gemini 3 Pro |
> | ----------------- | -------- | -------------- | ---------------- | ------------ |
> | Release Time      | Aug 2025 | Jun 2025       | Oct 2025         | Nov 2025     |
> | Knowledge Cut‑off | Oct 2024 | Jan 2025       | Feb 2025         | Jan 2025     |
>
> We also considered the risk that frequent model iterations by individual developers could lead to benchmark exploitation soon after release. To address this, we plan to operate a smaller private leaderboard updated monthly to detect and publicly report potential hacking attempts.
>
> To assess its reliability, we constructed two evaluation sets of 140 items using data from July and August 2025 and tested all models. As shown below for representative models, score differences were small and rankings remained consistent, confirming the private leaderboard’s effectiveness for contamination detection. Please see lines 501-513 in the new version for details.
>
> | Model     | GPT‑5 | GPT‑4.1 | InternVL 3.5‑24B | Gemini 2.5 Pro | Claude 3.5 Sonnet | Qwen2.5‑VL‑72B | MedGemma‑27B | LingShu‑32B | HuatuoGPT‑V‑7B |
> | --------- | ----- | ------- | ---------------- | -------------- | ----------------- | -------------- | ------------ | ----------- | -------------- |
> | H1 2025   | 35.5  | 31.8    | 34.4             | 33.7           | 27.3              | 17.2           | 15.6         | 20.1        | 9.2            |
> | July 2025 | 36.8  | 33.2    | 35.8             | 35.1           | 28.7              | 18.6           | 16.9         | 21.4        | 10.5           |
> | Aug 2025  | 34.1  | 30.4    | 33.0             | 32.3           | 26.0              | 15.8           | 14.2         | 18.7        | 8.0            |
>
> [1]. [LiveBench: A Challenging, Contamination-Limited LLM Benchmark](https://openreview.net/forum?id=sKYHBTAxVa)
>
> [2]. [LiveCodeBench: Holistic and Contamination Free Evaluation of Large Language Models for Code](https://arxiv.org/abs/2403.07974)

---

> ### Author Response · Authors · 2025-11-21
> **Reply to sxpY, Part 4**
>
> > Q1: How do you ensure the case reports selected are representative of disease distributions in real clinical practice? Does collecting data from PMC result in higher probability of rare cases?
>
> We appreciate your concern about the representativeness of case reports and the potential bias toward rare diseases. Patient populations vary widely across regions, hospital levels, and specialties, making it impossible to define a single “gold‑standard” distribution for all benchmarks. Our scope is guided by one of our core aims: capturing emerging medical knowledge, which has been doubling every 73 days since 2020 [1]. In this context, daily case report publications meaningfully reflect the distribution of new clinical insights and align with the goals of a *live* benchmark.
>
> While case reports often include rare conditions, the [NIH guideline](https://pmc.ncbi.nlm.nih.gov/articles/PMC5686928/) notes other common reasons for their publication, such as unexpected symptom correlations, unforeseen treatment events, or novel therapeutic approaches. Working with clinicians, we annotated each case for rarity (rare/unrare) and built a subset of more common diseases.
>
> We then compared model accuracy across rarity subsets to examine whether errors mainly stemmed from out‑of‑distribution rare cases. For most models, accuracy differences between the unrare subset and the full set were within ±5 percentage points. Stronger models are less affected by rarity, and some GPT variants perform better on rare cases. These results suggest that rarity has limited impact on scores and rankings. Please see lines 514-523 and 1547-1610 in the new version for details.
>
> | Model Name          | Rare (1181, 84%) | Unrare (226, 16%) | Δ(U,R) | OverAll |
> | ------------------- | ---------------- | ----------------- | ------ | ------- |
> | GPT-5               | 35.65%           | 35.05%            | -0.60% | 35.5%   |
> | O3                  | 35.99%           | 34.11%            | -1.88% | 35.7%   |
> | GPT-4.1             | 31.92%           | 30.84%            | -1.08% | 31.8%   |
> | GPT-4.1-mini        | 26.32%           | 25.00%            | -1.32% | 26.1%   |
> | GPT-4o              | 14.23%           | 17.76%            | 3.53%  | 14.7%   |
> | GPT-4o-mini         | 14.15%           | 19.34%            | 5.19%  | 14.8%   |
> | InternVL 3.5-241B   | 33.89%           | 37.12%            | 3.23%  | 34.4%   |
> | nternVL 3.5-38B     | 17.28%           | 21.78%            | 4.50%  | 18.0%   |
> | InternVL 3.5-8B     | 11.63%           | 16.48%            | 4.85%  | 12.4%   |
> | Gemini-2.5 Pro      | 33.36%           | 35.05%            | 1.69%  | 33.7%   |
> | Gemini-2.5 Flash    | 18.54%           | 20.56%            | 2.02%  | 19.0%   |
> | Gemini-2.0 Flash    | 20.61%           | 25.00%            | 4.39%  | 21.4%   |
> | Gemini-1.5 Pro      | 17.02%           | 20.28%            | 3.26%  | 17.7%   |
> | Claude-3.7 Sonnet   | 25.06%           | 29.91%            | 4.85%  | 25.6%   |
> | Claude-3.5 Sonnet   | 27.01%           | 29.44%            | 2.43%  | 27.3%   |
> | Claude-3.5 Haiku    | 14.92%           | 19.63%            | 4.71%  | 15.3%   |
> | LingShu-32B         | 20.00%           | 20.40%            | 0.40%  | 20.1%   |
> | LingShu-7B          | 15.24%           | 15.64%            | 0.40%  | 15.3%   |
> | GLM-4.1V-9B         | 17.56%           | 21.96%            | 4.40%  | 18.8%   |
> | MiMo-VL-7B-RL-2508  | 10.32%           | 14.49%            | 4.17%  | 10.9%   |
> | Qwen2.5-VL-72B      | 17.87%           | 21.96%            | 4.09%  | 18.6%   |
> | Qwen2.5-VL-32B      | 16.49%           | 21.50%            | 5.01%  | 17.2%   |
> | Qwen2.5-VL-7B       | 7.73%            | 8.41%             | 0.68%  | 7.9%    |
> | MedGemma-27B It     | 14.76%           | 20.04%            | 5.28%  | 15.6%   |
> | MedGemma-4B It      | 7.56%            | 7.48%             | -0.08% | 7.5%    |
> | HuatuoGPT-Vision-7B | 8.73%            | 13.08%            | 4.35%  | 9.2%    |
>
> [1] [Crises and Turnaround Management: Lessons Learned from Recovery of New Orleans and Tulane University Following Hurricane Katrina](https://pubmed.ncbi.nlm.nih.gov/30309439/#&gid=article-figures&pid=figure-2-uid-1)

---

> ### Author Response · Authors · 2025-11-21
> **Reply to sxpY, Part 5**
>
> > Q3: How does model performance correlate between LiveClin and real clinical decision-making tasks or other clinical benchmarks?
>
> We appreciate your question on the relationship between LiveClin and real clinical decision making as well as other clinical benchmarks. We compared representative models on LiveClin with seven established multimodal clinical benchmarks: OmniMedVQA (OM.VQA) [1], PMC‑VQA [2], VQA‑RAD [3], SLAKE [4], PathVQA [5], MedXpertQA (MedXQA) [6], and MMMU‑Med [7]. We then calculated Pearson and Spearman correlations between LiveClin accuracy and each benchmark.
>
> As shown in table below, LiveClin shows the strongest alignment with clinically oriented multimodal benchmarks such as MedXpertQA (ρ = 0.62, r = 0.48), which require integration of heterogeneous patient data and context‑specific decision making. In contrast, alignment with primarily single‑turn image‑centric VQA tasks such as OmniMedVQA and PMC‑VQA is much lower, reflecting their limited ability to assess complex reasoning. These results indicate that LiveClin captures core competencies relevant to real clinical decision making.
>
> | Model            | LiveClin (%) | OM.VQA | PMC-VQA | VQA-RAD | SLAKE | PathVQA | MedXQA   | MMMU-Med |
> | ---------------- | ------------ | ------ | ------- | ------- | ----- | ------- | -------- | -------- |
> | GPT-5            | 35.5         | 76.4   | 60.0    | 67.8    | 78.1  | 52.8    | 71.0     | 83.6     |
> | GPT-5-mini       | 35.7         | 70.9   | 57.6    | 66.3    | 76.1  | 52.4    | 60.1     | 80.5     |
> | GPT-4.1          | 31.8         | 75.5   | 55.2    | 65.0    | 72.2  | 55.5    | 45.2     | 75.2     |
> | GPT-4o           | 14.7         | 67.5   | 49.7    | 61.0    | 71.2  | 55.5    | 44.3     | 62.8     |
> | Gemini-2.5-Flash | 33.7         | 71.0   | 55.4    | 68.5    | 75.8  | 55.4    | 52.8     | 76.9     |
> | Qwen2.5-VL-7B    | 10.9         | 63.6   | 51.9    | 63.2    | 66.8  | 44.1    | 20.1     | 50.6     |
> | Qwen2.5-V-32B    | 17.2         | 68.2   | 54.5    | 71.8    | 71.2  | 41.9    | 25.2     | 59.6     |
> | InternVL3-8B     | 12.4         | 79.1   | 53.8    | 65.4    | 72.8  | 48.6    | 22.4     | 59.2     |
> | InternVL3-38B    | 18.0         | 79.8   | 56.6    | 65.4    | 72.7  | 51.0    | 25.2     | 65.2     |
> | MedGemma-27B     | 15.6         | —      | 49.5    | 62.3    | 74.9  | 44.4    | 33.9     | 56.2     |
> | MedGemma-4B      | 7.9          | 70.7   | 49.2    | 72.3    | 78.2  | 48.1    | 25.4     | 43.2     |
> | HuatuoGPT-V-7B   | 9.2          | 74.3   | 53.1    | 67.6    | 68.1  | 44.8    | 23.2     | 49.8     |
> | Lingshu-7B       | 20.1         | 82.9   | 56.3    | 67.9    | 83.1  | 61.9    | 26.7     | 54.0     |
> | Lingshu-32B      | 15.3         | 83.4   | 57.9    | 76.7    | 86.7  | 65.5    | 30.9     | 62.3     |
> | Pearson r        | —            | 0.17   | 0.45    | 0.39    | 0.43  | 0.46    | 0.48     | 0.36     |
> | Spearman ρ       | —            | 0.31   | 0.48    | 0.40    | 0.51  | 0.55    | **0.62** | 0.43     |
>
> To assess alignment with actual clinical decision making, we measured physician accuracy on 100 randomly sampled LiveClin cases. As shown in table below, Chief Physicians achieved higher scores than Attending Physicians, and both groups performed better than most models. Only GPT‑5 and O3 slightly exceeded Attending Physicians but still fell short of Chief Physicians. This shows that LiveClin is sensitive to differences in expertise and reflects the difficulty of real clinical decision making, while most current medical AI systems remain far from managing full clinical pathways. Please see lines 324-338 and 348-350 in the new version for details.
>
> | Model/Physician | Chief Physician | Attending Physician | GPT‑5 | O3   | GPT‑4.1 | Gemini 2.5 Pro | Claude 3.7 Sonnet |
> | --------------- | --------------- | ------------------- | ----- | ---- | ------- | -------------- | ----------------- |
> | ACC             | 37.8            | 34.6                | 35.5  | 35.7 | 31.8    | 33.7           | 25.6              |
>
> These results demonstrate that LiveClin correlates well with complex clinically grounded benchmarks and with physician performance differences, supporting its validity as a measure of real clinical decision making.
>
> [1]. [OmniMedVQA: A New Large-Scale Comprehensive Evaluation Benchmark for Medical LVLM](https://arxiv.org/abs/2402.09181)
>
> [2]. [PMC-VQA: Visual Instruction Tuning for Medical Visual Question Answering](https://arxiv.org/abs/2305.10415)
>
> [3]. [A dataset of clinically generated visual questions and answers about radiology images](https://www.nature.com/articles/sdata2018251)
>
> [4]. SLAKE: A Semantically-Labeled Knowledge-Enhanced Dataset for Medical Visual Question Answering
>
> [5]. PathVQA: 30000+ Questions for Medical Visual Question Answering
>
> [6]. MedXpertQA: Benchmarking Expert-Level Medical Reasoning and Understanding
>
> [7]. MMMU: A Massive Multi-discipline Multimodal Understanding and Reasoning Benchmark for Expert AGI

---

> > ### Author Response · Authors · 2025-11-26
> > **Follow-Up on Review and Feedback**
> >
> > Dear Reviewer **sxpY**,
> >
> > We hope this message finds you well.
> >
> > We have carefully addressed all your questions and concerns, including conducting additional experiments as requested, and have provided detailed responses in the rebuttal.
> >
> > As the rebuttal deadline is approaching, we would deeply appreciate it if you could share your updated thoughts based on the rebuttal and paper revision, or do not hesitate to let us know if you have additional questions, and we will respond promptly.
> >
> > Thank you again for your thoughtful review and your invaluable contributions to the quality of this paper.
> >
> > Kind regards,
> >
> > Paper 19066 Authors

---

> > > ### Comment · Reviewer_sxpY · 2025-11-28
> > >
> > > The authors have addressed most of my questions. I believe that incorporating these into the updated draft would greatly strengthen the paper. I have raised my scores.

---

### Author Response · Authors · 2025-11-21
**General Response**

We thank all reviewers for their detailed feedback and constructive suggestions across multiple aspects of our work. We greatly appreciate that reviewers recognized the uniqueness, rigor, and clinical relevance of **LiveClin**, a clinician‑validated, multimodal benchmark designed to be contamination‑resistant, regularly updated, and capable of simulating the entire patient care pathway.

**Key additions and clarifications based on reviewer feedback:**

1. **Physician Baselines**: Accuracy for Attending and Chief Physicians on 100 cases shows physicians outperform almost all models (Q2, 1rMm, y36J). Details are shown in lines 324–338, 348–350.
2. **Biannual Update Plan**: Detailed update workflow, resource estimates (USD 53.5K, 11.3 days per cycle), and full retesting commitment; confirmed sustainability within current team capacity (W3, sxpY, y36J, KBAZ). Details are shown in lines 501–506.
3. **Contamination Controls**:  Quantified model release vs. data cut‑off gaps (6–8 months), described private leaderboard guardrail; demonstrated stable scores and rankings across monthly sets (W4/Q2, sxpY, y36J). Details are shown in lines 501–513.
4. **Case Report Rarity Analysis**: Annotated cases by rarity and compared performance across rare/unrare subsets; found ≤5 pp differences for most models, with stronger models less affected, mitigating bias concerns (Q1, sxpY, 1rMm, KBAZ). Details are shown in lines 514–523 and 1547–1610.
5. **MCQ Design Rationale and Mitigation**:  Justified MCQs for objectivity and scalability; added scenario design and 10 distractors (W4, 1rMm).

**Overall**, these changes strengthen the paper’s empirical grounding, transparency, and operational details, while directly addressing each reviewer’s concerns. LiveClin occupies a distinctive position in the research ecosystem:

- A *live*, contamination‑resistant, clinically grounded benchmark
- Scalable yet rigorously physician‑validated through an AI–human collaborative pipeline
- Covering the full clinical pathway with aligned multimodal reasoning tasks
- Providing both granular taxonomy analysis and correlation with real‑world benchmarks and physician performance

We thank the reviewers again for their constructive insights, which have helped us significantly improve the clarity, completeness, and evidential support of this work. We look forward to further advancing contamination‑resistant, clinically relevant evaluation for medical AI.

---

### Author Response · Authors · 2025-11-30
**Rebuttal Summary for New-Round AC**

We sincerely appreciate the additional time and effort you have dedicated to supporting our community during such a challenging moment. Acknowledging the constraints on your time, we have prepared the following concise summary of the key interactions during the rebuttal period, alongside paper and review excerpts, to facilitate the understanding and save time. Thanks again for your responsible and thoughtful efforts.

**Timeline of Reviewer Responses and Score Changes**

| Reviewer | Before Leak Event                                            | After Leak Event                                             |
| -------- | ------------------------------------------------------------ | ------------------------------------------------------------ |
| **sxpY** | ***Overall score***: 4.    *Response*: None              | ***Response***: “The authors have addressed most of my questions. **I have raised my scores.**”  However, as scores had already been frozen, the intended updated score is unknown. |
| **1rMm** | ***Overall score***: **6 to 8**.    *Response*: “Thank you for your detailed response which addressed most of my concerns. I have raised my score accordingly.” *On Nov 24* | No update                                                    |
| **y36J** | ***Overall score***: 4.    *Response*: None              | No update                                                    |
| **KBAZ** | ***Overall score***: **4 to 8**.     *Response*: “My concerns have been addressed and LiveClin is an interesting contribution for medical LLM evaluation.” *On Nov 23* | No update                                                    |

We thank all reviewers for their detailed feedback and constructive suggestions across multiple aspects of our work. We greatly appreciate that reviewers recognized the uniqueness, rigor, and clinical relevance of **LiveClin**, a clinician‑validated, multimodal benchmark designed to be contamination‑resistant, regularly updated, and capable of simulating the entire patient care pathway.

**Key additions and clarifications based on reviewer feedback:**

1. **Physician Baselines**: Accuracy for Attending and Chief Physicians on 100 cases shows physicians outperform almost all models (Q2, 1rMm, y36J). Details are shown in lines 324–338, 348–350.
2. **Biannual Update Plan**: Detailed update workflow, resource estimates (USD 53.5K, 11.3 days per cycle), and full retesting commitment; confirmed sustainability within current team capacity (W3, sxpY, y36J, KBAZ). Details are shown in lines 501–506.
3. **Contamination Controls**:  Quantified model release vs. data cut‑off gaps (6–8 months), described private leaderboard guardrail; demonstrated stable scores and rankings across monthly sets (W4/Q2, sxpY, y36J). Details are shown in lines 501–513.
4. **Case Report Rarity Analysis**: Annotated cases by rarity and compared performance across rare/unrare subsets; found ≤5 pp differences for most models, with stronger models less affected, mitigating bias concerns (Q1, sxpY, 1rMm, KBAZ). Details are shown in lines 514–523 and 1547–1610.
5. **MCQ Design Rationale and Mitigation**:  Justified MCQs for objectivity and scalability; added scenario design and 10 distractors (W4, 1rMm).
6. **Inter‑Annotator Agreement Data**:  Reported 8.7% disagreement rate between physicians, 90% resolved in one loop; all resolved in ≤2 loops, indicating high agreement (W2, y36J).  Details are shown in lines 260–261 and 1129–1138.

**Overall**, these changes strengthen the paper’s empirical grounding, transparency, and operational details, while directly addressing each reviewer’s concerns. LiveClin occupies a distinctive position in the research ecosystem:

- A *live*, contamination‑resistant, clinically grounded benchmark
- Scalable yet rigorously physician‑validated through an AI–human collaborative pipeline
- Covering the full clinical pathway with aligned multimodal reasoning tasks
- Providing both granular taxonomy analysis and correlation with real‑world benchmarks and physician performance

We thank the reviewers again for their constructive insights, which have helped us significantly improve the clarity, completeness, and evidential support of this work. We look forward to further advancing contamination‑resistant, clinically relevant evaluation for medical AI.

---

### Meta-Review · Area_Chair_gGg8 · 2025-12-23

**Summary:**

**Summary of contributions**: Live Clin proposes a live benchmark for clinical reasoning that is robust to leakage and represents the complexity of clinical reasoning. The framework includes bi-annual update strategies. To generate the benchmark, the paper proposes a three-stage approach using the open subset of PubMed. The first stage curates peer-reviewed cases and extracts the stages of clinical reasoning and decision-making. In the second stage, a generator creates questions that are rated by a critic for accuracy and cognitive complexity, including a refinement stage to improve on both axes. Ablation studies confirm the critic is useful to focus case-generation on cross-model/cross-step reasoning and retaining authentic but potentially incomplete information in case-generation. Quality checks are performed in stage-3 constituting an LLM-as-a-judge for factual validation and solvability. This is followed by physician screening to ensure iterative refinement on final diagnosis through  consensus building loop.

Major specialties are represented in different proportion and overall 1,772.18 person hours were spent to create one version of the benchmark. The benchmark is used to evaluate proprietary, open-source, and medicine focused LLMs. with results demonstrating that proprietary LLMs demonstrate strong performance beating attendings but not surpassing chief physicians especially when complex reasoning is necessary. Significant performance discrepancies are observed across specialties and modalities. Where logical rule-based application is necessary, LLMs excel, but when complex clinical reasoning is necessary, current llms do not perform well. Finally the paper demonstrates a workflow to prevent contamination, and account for bias while enabling impact assessment.

**Reviewer comments**:
1. Reviewers raised concerns about whether using published cases would emphasize rare conditions and cases over clinically representative data.

2. Concerns were raised that emphasis on zero-shot evaluation may be confounded with adapting to formatting rather than clinical content

3. Contextualize better what use-cases justify using this as the only evaluation benchmark.

4. Since the cost of generating the benchmark in one round was about 43K USD, the sustainability was unclear

5. Comparison and contextualization to other decontamination strategies and whether that saves costs.

6. Reliance on multiple choice QA is a limitation that needs to be better acknowledged

**Reviewer Concerns:**

Authors have conducted extensive experiments and updated the manuscript to reflect their improvements. In particular, reviewer concerns about emphasis on rare cases due to reliance on published case-studies was acknowledged, and further assessment conducted on distribution of tasks, as well as clinician rating on rarity of cases was sought with quantified inter-rater reliance.

Second, few-shot experiments were added.

Authors added contextual discussion compared to prior work,

Justified the logistics and mechanism of maintaining the live benchmark

Justified the choice of multiple choice style question answering use in the framework

Added additional discussion on inter-rater reliability for the proposed benchmark/refinements in the manuscript as well as to the response

**Reviewer Scores:**

Reviewers note sxpY mentioned in text that they would like to raise scores.
Reviewer 1rMm also noted that they would like to raise scores.
Finally reviewer KBAZ also noted that they would like to raise scores.

Reviewer y36J did not respond to author rebuttal, however, their rebuttal seems to clarify many issues raised by y36J.

Overall I think the authors comprehensively address many of the concerns, and I suggest authors explicitly discuss limitations brought up by the reviewers in the final camera-ready. The presentation of the updated experiments should also be improved in the camera-ready.

---

### Decision · Program_Chairs · 2026-01-26

Accept (Poster)